



# Simulation Analysis of Local Land Atmosphere Coupling in Rainy Season over a Typical Underlying Surface in the Tibetan Plateau

Genhou SUN[1,2], Zeyong HU[3], Yaoming MA[4,5], Zhipeng XIE[4], Jiemin WANG[3], Song YANG[1,2]

[1]School of Atmospheric Sciences and Guangdong Province Key Laboratory for Climate Change and Natural Disaster Studies,
Sun Yat-sen University, Guangzhou, China

[2] Southern Marine Science and Engineering Guangdong Laboratory (Zhuhai), Zhuhai, China

[3]Northwest Institute of Eco-Environment and Resources, Chinese Academy of Sciences, Lanzhou, China

[4]Institute of Tibetan Plateau, Chinese Academy of Sciences, Beijing, China

[5]CAS Center for Excellence in Tibetan Plateau Earth Sciences, Beijing, China

*Correspondence to*: Genhou SUN (sungh6@mail.sysu.edu.cn); Yaoming Ma (ymma@itpcas.ac.cn)

**Abstract.** The Local land atmosphere coupling (LoCo) focuses on the interactions between soil conditions, surface fluxes, planetary boundary layer (PBL) growth, and the formations of convective clouds and precipitations. Study of LoCo over the Tibetan Plateau (TP) is of great significance for understanding TP's role in the "Asian Water Tower". A series of real-case simulations using the Weather Research and Forecasting Model (WRF) with different combinations of land surface models
(LSM) schemes and PBL schemes has been carried out to investigate the LoCo characteristics over a typical underlying surface in the central TP in rainy season. The LoCo characteristics in the study area are analyzed by applying a mixing diagram to the simulation results. The analysis indicates that the WRF simulations using the Noah with BouLac, MYNN, and YSU produce closer results to the observation in terms of curves of $Cp*\theta$ and $Lv*q$, surface fluxes ($H_{sfc}$ and $LE_{sfc}$), entrainment fluxes ($H_{ent}$ and $LE_{ent}$) at site BJ/Nagqu than those using the CLM with BouLac, MYNN, and YSU. The frequency distributions of $H_{sfc}$,
$LE_{sfc}$, $H_{ent}$, and $LE_{ent}$ in the study area confirm this result. The spatial distributions of simulated $H_{sfc}$, $LE_{sfc}$, $H_{ent}$, and $LE_{ent}$ using WRF with Noah and BouLac suggest that the spatial distributions of $H_{sfc}$ and $LE_{sfc}$ in the study area are consistent with that of soil moisture, but the spatial distributions of $H_{ent}$ and $LE_{ent}$ are quite different from that of soil moisture. A close examination of the relationship between entrainment fluxes and cloud water content (QCloud) reveals that the grids with small $H_{ent}$ and large $LE_{ent}$ tend to have high QCloud and $H_{sfc}$, suggesting that high $H_{sfc}$ is conductive to convective cloud formation, which
leads to small $H_{ent}$ and large $LE_{ent}$. Sensitivity analysis of LoCo to the soil moisture at site BJ/Nagqu indicates that on a sunny day, an increase in soil moisture leads to an increase in $LE_{sfc}$ but decreases in $H_{sfc}$, $H_{ent}$, and $LE_{ent}$. The sensitivity of the relationship between simulated maximum daytime PBL height (PBLH) and mean daytime evapotranspiration (EF) in the study area to soil moisture indicates that the rate at which the maximum daytime PBLH decreases with the mean EF increase as the initial soil moisture goes up. The analysis of simulated $H_{sfc}$, $LE_{sfc}$, $H_{ent}$, and $LE_{ent}$ under different soil moisture conditions
reveals that the frequency of $H_{ent}$ ranging from 80 to 240 W/m² and the frequency of $LE_{ent}$ ranging from -240 to -90 W/m² both increase as the initial soil moisture increases. Coupled with the changes in QCloud, the changes in $H_{ent}$ and $LE_{ent}$ as the initial soil moisture increases indicate that the rise in soil moisture leads to an increase in the cloud amount but a decrease in QCloud.

# 1. Introduction





With an average altitude of over 4000 m, the Tibetan Plateau (TP) is also known as the "Asian Water Tower" because it is home to many well-known rivers like the Yangzi River, the Yellow River, the Mekong River, the Yalung Tsangpo River, the Indian River, and the Ganges River (Lu et al., 2005; Immerzeel et al., 2010; Pithan et al., 2014). Local land atmosphere coupling (LoCo) (Santanello et al., 2009, 2011, 2018) focuses on the interactions between land surface states, surface fluxes, planetary boundary layer (PBL) development and the formations of convective clouds and precipitations, which are closely

linked to the relevant processes of the Water Tower. Therefore, studying the LoCo over the TP is of great significance for understanding the TP's role in Asian Water Tower.

      LoCo is defined by Santanello (Santanello et al.,,, 2011) as the interactions among the land surface state, surface fluxes, PBL development, and the developments of convective clouds and precipitation. The processes of LoCo can be simplified four individual processes: 1) the responses of surface fluxes to land surface states, 2) the responses of PBL evolutions to surface

fluxes, 3) the responses of entrainment fluxes, 4) the collective feedback of the free atmosphere on PBL thermodynamics. Therefore, LoCo analysis can help us understand land-atmospheric interactions from meteorological and climatological perspectives. These processes are highly nonlinear, making it difficult to quantitatively understand Loco in a reliable way. Therefore, a couple of metrics, which try to produce quantitative understanding of LoCo, have been developed (Santanello et al., 2018). A framework called "mixing diagram" is one of the metrics (Santanello et al., 2009, 2011, 2018) which analyzes

the PBL thermodynamics using the 2-m potential temperature and humidity, surface fluxes and PBL height (PBLH). Santanello (2009) constructed a diagnostic framework "mixing diagram" to investigate quantitatively the relationship between soil moisture and PBL evolution, and PBL energy budget. Then, they furthered this study in Santanello (2011) by combining the mixing diagram with the equivalent potential temperature and the level of lifting condensation to examine the possible role of soil moisture in the formation and development of convective clouds in the Southern Great Plains. The positive and negative

feedbacks of LoCo on the formations of convective clouds and precipitation were identified. Santanello (2019) investigated the possible impact of initial soil impact on the numerical weather prediction (NWP) and stated that both initial soil moisture and spatial resolution may have profound impacts on the spread of the NWP. The LoCo analysis using the mixing diagram show a great potential to advance our understanding of the interactions between soil conditions and surface fluxes, PBL growth and the formations of convective clouds.

In the last few decades, our understanding of the individual processes of LoCo in TP has been greatly advanced because a series of scientific programs have been carried out in the TP. Based on the observed datasets from the stations established in the TP, the spatiotemporal distribution of soil temperature and soil moisture in TP has been obtained (Yang et al.,, 2013; Su et al.,, 2013; Qin et al.,, 2013; Zhao et al., 2014), the temporal variations of surface fluxes over different underlying surfaces have been quantitively analyzed (Yao et al 2008; Sun et al., 2016, 2019; Wang et al., 2017; Xin et al., 2018). The spatial distribution

of surface fluxes over the whole TP have been gained based on in-sit measurements and satellite images (Ma et al., 2005, 2009, 2014) and long-term variations of surface fluxes have investigated (Yang et al., 2011, 2014; Han et al., 2017). The profiles of temperature, humidity, and wind speed obtained using radiosonde in Nagqu (Li et al., 2004), NamCo (Lv et al., 2008), Madoi (Li et al., 2017) and Qomulangma (Chen et al., 2008; Li et al., 2011) were used to investigate PBL evolution at different





stations in seasons over the TP. These studies suggested that the PBL in the rainy and dry seasons exhibited distinct daily evolutions and seasonal variations due to the diurnal and seasonal variations of strong surface heating. Properties of clouds and precipitation have been gained thanks to the radar and lidar measurements over the TP. Liu (2015) studied the clouds and precipitation at Nagqu using one-year radar and lidar observations; they stated that the cloud amount and depth in the rainy season exhibited distinct diurnal variation and that strong updrafts existed during the formation of cumuli. Chang (2016) studied the characteristics of convective clouds in summer based on radar and lidar measurements and found that surface heating played an important role in diurnal variations of convective clouds and precipitation. The simulation analysis of mechanism convective precipitation events in TP (Chen et al., 2019) stressed the important role of surface heating in the formation of convective precipitations in the east and west part of TP in summer. These studies have greatly advanced our understanding of the relevant processes of LoCo in the TP based on in-situ measurements, remote sensing images, and radar and lidar measurements; they highlighted possible connections among these individual processes. However, the underlying mechanisms are still unclear. In this study, we will investigate the possible roles of land surface conditions in convective cloud formation on a sunny day of the rainy reason in terms of LoCo characteristics.

Site BJ of Nagqu Station for Plateau Climate and Environment, Chinese Academy of Sciences (site BJ/Nagqu, 31.37°N, 91.90°E, 4509 m) is located in the central TP and has been chosen by a series of scientific programs as the key station to carry out their field measurements (Ma et al., 2008, 2014a; Zhao et al., 2018, 2019). In this study, an area of 150 km by 150 km with Site BJ/Nagqu in the center was chosen as our study area. One important reason for taking this region as our study area is that there was an intense observation period for a scientific program in 2011 at Site BJ/Nagqu (Ma et al., 2014a). Those observation data will be used in this study. Another reason is that the vegetation types in the study area are alpine grassland and alpine wetland, which are very common in the TP. The analysis based on the measurements at Site BJ/Nagqu thus can provide valuable information to understand the land atmosphere coupling over the TP. The paper is organized as follows. Section 2 introduces the study area, the observation data, and setups of simulations in this study. Section 3 presents the LoCo characteristics in the study area on a sunny day in terms of PBL energy budget and the possible impact of soil condition on shallow convective clouds. Section 4 present a discussion based on the analysis above. Section 5 shows the conclusions and the future plans.

## 2. Study area and methodology

### 2.1 Study area

Simulations using three-nested domains (Fig. 1) are carried out. The innermost domain is a region of 150 km*150 km with Site BJ/Nagqu in the center (Fig. 1 c)). The altitude in the study area varies from 4400 to 5800 m. The central area is relatively flat, but there are mountains in the south and west parts of the study area. Based on the observation results of the Nagqu Meteorological Station of the Chinese Meteorological Administration (http://data.cma.cn/) from 1980 to 2016, the multi-year monthly mean temperature varies from -11.81 to 9.59°C, and the multi-year monthly mean rainfall varies from





2.16 to 109.69 mm. The climate from July to September is wet and warm, with frequent convective clouds and precipitation due to the strong surface heating and to the abundant water vapor brought in by the South Asian monsoon.

A scientific experiment was carried out at Site BJ/Nagqu in July and August 2011. The measurement instruments of

this experiment included one automatic weather station (AWS), which provided the measurements of air temperature and humidity at both 1.0 and 8.4 m, wind direction and speed at 10.0 m, surface pressure, the instruments for the 4-layer soil temperature and moisture (5, 10, 20, and 40 cm), one eddy covariance system for surface fluxes and radiosonde measurements for the profiles of temperature, humidity, and wind. Profile data of temperature, humidity, and wind in the daytime were collected in this campaign. The profile data coupled with surface measurements make it possible to carry

out LoCo simulation analysis in the central TP in the rainy season.

## 2.2 WRF model setup

The WRF is a state-of-the-art mesoscale numerical simulation system and has been widely used in atmospheric research and forecasting applications. Developed based on the fifth-generation Mesoscale Model (MM5), WRF has a Eulerian mass dynamic core and offers a flexible platform, which can combine various schemes of radiation, land surface

processes, PBL, and microphysics. The system can produce simulations based on real or idealized atmospheric conditions.

WRF 3.6.1 is used in this study to simulate LoCo interactions in the rainy season in the TP. There are 99*99 grid cells with a grid resolution of 0.135° *0.135° in the outermost domain, and 99*99 grid cells with a grid resolution of 0.015°*0.015° in the innermost domain. The vertical levels in the simulations are 39 from the surface to 200 hPa. The

forcing data for the simulations are the ERA-Interim, with a spatial resolution of 0.75°*0.75° and a temporal resolution of 6 h. To carry out the real-case simulations, the initial soil moisture was modified based on in-situ measurements at Site BJ/Nagqu and the leaf area index (LAI) of the MODIS. This is based on the fact that the soil moisture and LAI show similar variation in the rainy season and that the soil moisture in the study from the ERA-Interim is too homogeneous, which is far from the reality. First, the relationship between LAI and soil moisture at 5 cm was derived based on the

observed 5-cm soil moisture data at Site BJ/Nagqu and LAI product in the rainy season in 2011 (Fig. 2 (a)). The derived relationship between LAI and soil moisture was applied to the LAI product of the study area to obtain a realistic soil moisture for the simulations. Note that there are four layers of soil moisture in Noah (10, 40, 100, 160 cm) and 10 layers of soil moisture in CLM (1.8, 4.5, 9.06, 16.6, 28.9, 49.3, 82.9, 138.3, 229.6, and 343.3 cm). We found the variation of soil moisture in the ERA-Interim from 40-cm depth to the top shows very small changes. We thus modified the soil from

40-cm depth to the top by applying the relationship between soil moisture at 5 cm and LAI (Fig. 2 b)).

The setups of WRF simulation in this study are as follows: WSM 5-class scheme for microphysics, RRTM scheme for shortwave, Duhia scheme for longwave radiation, and Kain-Fritsch (new Eta) scheme for convective cloud parameterization only in the outermost domain. The land surface process schemes used here are Noah (Ek et al., 2003) and CLM4 (Oleson et





al., 2010; Lawrence et al., 2010). The PBL schemes used in this study are Yonsei University (YSU; Hong et al., 2006), Mellor-
Yamada Nakanishi and Niino Level-2.5 PBL (MYNN; Nakanishi et al., 2001, 2004), and BouLac (Bougeault et al., 1989).
Simulations using WRF with different combinations of LSMs and PBL schemes are carried out to study LoCo characteristics
over a typical underlying surface in the central TP.

To investigate the sensitivity of LoCo characteristics to soil moisture, the simulations using WRF Noah-BouLac with
different initial soil moisture conditions are carried out. The initial soil moisture conditions in the study area for the sensitivity
simulations are set to real soil moisture minus 0.05 (real SM–0.05), real soil moisture (real SM), real soil moisture plus 0.05
(real SM+0.05), and real soil moisture plus 0.1 (real SM + 0.1).

All the simulations run for 36 h. The first 6 h of the simulation is for the spin-up and is not analyzed.

### 2.2.1 Land surface models

Both Noah and CLM4 can simulate surface sensible heat fluxes, latent heat fluxes, and soil states, but they are different
in specific parameterizations and representations of soil and vegetation properties and of physics. Noah calculates
moisture and heat transports using four layers of soil temperature and moisture in each grid cell. CLM4 calculates the
water and heat transports in the soil using 10 layers of soil temperature and soil moisture in each grid cell. Noah
determines the vegetation parameters (such as height, coverage, and density) using the Global Forecasting System (GFS),
while CLM4 has five land-use types that determine vegetation conditions through a look-up table.

**2.2.2 PBL schemes**

There are several options for PBL schemes in WRF 3.6.1. We choose YSU, MYNN, and BouLac schemes in this study.
YSU is a nonlocal PBL scheme, which uses the K-profile method to parameterize the turbulent mixing in the convective
boundary layer and takes the non-local mixing by convective large eddies into consideration. Besides, YSU has an
explicit treatment of entrainment fluxes at the top of the PBL. PBLH in the YSU scheme is determined using the bulk
Richardson number. MYNN is a one-and-a-half order, local closure scheme, and the turbulent fluxes of any adiabatically
conserved variable are calculated based on the gradients of their mean values at adjacent levels only. BouLac is also a
one-and-a-half order, local closure scheme, and needs one additional prognostic option to predict turbulence kinetic
energy (TKE).

### 2.3 Mixing diagram

The mixing diagram was first proposed by Betts (1984, 1992) in order to quantify the heat and moisture fluxes transported
into the PBL in the daytime. The mixing diagram quantifies the heat and water into the PBL from the surface and from the
entrainment based on the daytime evolutions of 2-m potential temperature and humidity, PBLH, and surface sensible and latent
heat fluxes ($H_{sfc}$ and $LE_{sfc}$) at the daytime. Then Santanello (2009, 2011) furthered this study by investigating the PBL
thermodynamics and the possible influence of surface heating on cloud formations. In this study, the mixing diagram was used
to explore the PBL thermodynamics and the role of surface heating in convective cloud formations. The observed and simulated
$T_2$, $q_{2m}$, $H_{sfc}$, $LE_{sfc}$, and PBLH from 08:00 to 17:00 (Beijing standard time) on August 7, 2011 were used in this study. The



reasons for why the data from 08:00 to 17:00 was used in the analysis are because the underlying surface starts to influence the PBL at 08:00 and the PBLH usually reaches its maximum at 17:00. The mixing diagram analysis based on observed $T_2$, $q_{2m}$, $H_{sfc}$, $LE_{sfc}$, and PBLH on August 07, 2011 is shown in Fig. 3. The black line is the evolutions of $Cp*\theta$ and $Lv*q$ in the daytime, and the red and purple lines are vectors for surface fluxes and entrainment fluxes. The point $(q_m, T_m)$ in Fig. 3 is the PBL condition caused by the heat and water vapor from the surface only. The equations for $q_m$, $T_m$, $H_{ent}$, and $LE_{ent}$ are listed as follows:

$$\overline{H_{sfc}} = Cp *(T_m - T_s)* (\rho_m*\overline{PBLH})/\Delta t \tag{1}$$

$$\overline{LE_{sfc}} = Lv*(q_m - q_s) *(\rho_m*\overline{PBLH})/\Delta t \tag{2}$$

$$\overline{H_{ent}} = Cp *(T_f - T_m)* (\rho_m*\overline{PBLH}) /\Delta t \tag{3}$$

$$\overline{LE_{ent}} = Lv* (q_f - q_m) *(\rho_m*\overline{PBLH} ) /\Delta t \tag{4}$$

where $T_s$ and $q_s$ are the potential temperature and specific humidity at 08:00, respectively; $T_f$ and $q_f$ are the potential temperature and specific humidity at 17:00, respectively; Cp is the air specific heat capacity constant (1004.7 J(kg*K)); Lv is the constant for latent heat of vaporization ($2.5*10^6$ J(kg*K)); $\rho_m$ is air density (kg/m³), $\Delta t$ is the seconds from 08:00 to 17:00.

## 2.4 Other variables

PBLH is an important variable for the mixing diagram. The observed PBLH in this study was determined using a potential temperature gradient method from the profiles of temperature obtained by radiosonde measurements.

Equivalent potential temperature ($\theta_e$) is the temperature a sample of air would have if all its moisture were condensed by a pseudo-adiabatic process and the sample then brought dry-adiabatically back to 10000 hPa. $\theta_e$ can be calculated using the following equation:

$$\theta_e = \left(T + \frac{Lv}{C_{pd}} * r\right) * (\frac{P0}{p})^{\frac{R_d}{C_{pd}}} \tag{5}$$

where $T$ is absolute temperature (K); P0 is pressure at sea level (1000 hPa); P is surface pressure (hPa); r is mixing ratio of water vapor (kg/kg); $C_{pd}$ is specific heat of dry air at constant pressure (1005.7 J/(kg*K)); and $R_d$ is specific gas constant for air (287.04 J/(kg*K)).

## 3. Result analysis

## 3.1 Mixing analysis at Site BJ/Nagqu

The mixing diagrams of August 7, 2011 based on the simulations using Noah with BouLac, MYNN, or YSU are shown in Fig. 4. The curves of simulated $Cp*\theta$ and $Lv*q$ using Noah with different PBL schemes show similar variations with differences in values. The simulated increases in $Cp*\theta$ using Noah with the three PBL schemes are similar, from about 320 to 335 kJ/kg. The changes in $Lv*q$ using Noah with the three PBL schemes are different. The increases in simulated $Lv*q$ using Noah with BouLac, MYNN, and YSU are 0.94, 0.50, and 2.48 kJ/kg, respectively. The $H_{sfc}$ simulated using Noah with BouLac,





MYNN, and YSU are 144.12, 162.95, and 145.83 W/m$^2$, respectively. The $LE_{sfc}$ simulated using Noah with BouLac, MYNN,

and YSU are 235.86, 218.98, and 236.44 W/m$^2$, respectively. The simulated $H_{ent}$ is positive and larger than $H_{sfc}$, and the simulated $LE_{ent}$ values are negative. This suggests that significant heat and dry air are entrained into the PBL. Compared to the observed fluxes, the simulations using Noah with different PBL schemes produce larger $H_{sfc}$ and similar $LE_{sfc}$, and also produce smaller $H_{ent}$ and similar $LE_{ent}$.

$\theta_e$ is a thermal conservative variable, which is useful to quantify the changes in thermodynamics in the PBL due to surface

heating. The increases in simulated $\theta_e$ using Noah with BouLac, MYNN, and YSU are similar, indicating that the atmosphere becomes unstable due to the surface heating. The simulated relative humidity (RH) in the three cases decreases first from 80% to 60% at a slow rate, then decreases to below 40% at a fast rate. Therefore, the PBL becomes dryer and more unstable on the sunny day.

The mixing diagrams of August 7, 2011 based on simulations using CLM with BouLac, MYNN, and YSU are shown in Fig

5. The simulated curves using CLM with the three PBL schemes are different from those simulated using Noah. The Cp*$\theta$ simulated using CLM with different PBL schemes increases from 315.0 to about 337.0 kJ/kg. The Lv*q values simulated using CLM-BouLac and CLM-YSU increase from 10.0 to about 12.3 kJ/kg, while the Lv*q simulated using CLM with MYNN decreases from 10.13 to 9.5 kJ/kg. The simulated $H_{sfc}$ and $LE_{sfc}$ by CLM with different PBL schemes are similar, which may be one of the reasons for the similarity in these curves. The simulated $H_{ent}$ values using CLM with Boulac and YSU are than

the observed, while the simulated $H_{ent}$ using CLM with MYNN is smaller than the observed. All the simulated $LE_{ent}$ values using CLM with BouLac, MYNN, and YSU are smaller than the observed.

The simulated $\theta_e$ increases from about 324.83 to 345.21 K for CLM-MYNN, and from 324.56 to 348.80 K for CLM-BouLac and CLM-YSU. This indicates that the simulated PBL becomes more unstable in the CLM cases than it does in the Noah cases. The RH in PBL decreases from about 100% to about 30%. The simulated PBL using CLM is warmer and dryer than that

simulated by Noah.

## 3.2 PBL energy budget analysis at Site BJ/Nagqu

The PBL energy budgets at Site BJ/Nagqu simulated using different LSMs and PBL schemes are shown in Fig. 6. The simulated surface fluxes by Noah and CLM are different, which leads to differences in entrainment fluxes and the total fluxes

($H_{tot}$ and $LE_{tot}$). Both $H_{ent}$ and $H_{tot}$ simulated using CLM with BouLac, MYNN, and YSU are higher than those simulated by Noah, which means more warm air is entrained into PBL in the CLM simulations than it does in the Noah simulations. Both $LE_{ent}$ and $LE_{tot}$ simulated using Noah with BouLac, MYNN, and YSU are smaller than those simulated by CLM, suggesting that there is more dry air entrained into PBL in the Noah simulations. The Noah simulations with BouLac, MYNN, and YSU produce much larger $H_{sfc}$ and slightly larger $LE_{sfc}$ than the observed. The CLM simulations with BouLac, MYNN, and YSU

produce larger $H_{sfc}$ but smaller $LE_{sfc}$ than the observed. The $H_{ent}$ simulated using Noah is smaller than the observed, while the $H_{ent}$ values simulated using CLM with BouLac and YSU are larger than the observed. Compared to the observation, most simulated $LE_{ent}$ values are larger than the observed, indicating that less dry air is entrained into PBL in the simulations.





### 3.3 PBL energy budget analysis in the study area

The frequencies of PBL energy budget components in the study area simulated using WRF with different combinations of

LSMs and PBL schemes are shown in Fig. 7. There are apparent differences in the frequencies of simulated PBL budget components, suggesting discrepancies in simulated LoCo characteristics in the rainy season in the TP using WRF with different combinations of LSMs and PBL schemes.

The $H_{sfc}$ values simulated using CLM with BouLac, MYNN, and YSU vary in narrow ranges. The mean $H_{sfc}$ simulated using CLM is larger than that using Noah with BouLac and YSU but smaller than that using Noah with MYNN. The simulated $LE_{sfc}$

using CLM with BouLac, MYNN, and YSU show little differences and vary in narrower ranges than those simulated using Noah. CLM with BouLac, MYNN, or YSU produces stronger $H_{ent}$ and $LE_{ent}$ than Noah does. This suggests that more warm air and less dry air are entrained into PBL in the CLM simulations than in the Noah simulations. Consequently, CLM produces stronger $H_{tot}$ and $LE_{tot}$ than Noah does.

Here, we take the spatial distributions of PBL energy budget components simulated using Noah with BouLac as an example

to investigate the possible relationship between soil moisture and LoCo characteristics in the study area. The spatial distribution of the PBL energy budget simulated using Noah with BouLac is shown in Fig. 8. The heterogeneity in the spatial distributions of $H_{sfc}$, $LE_{sfc}$, $H_{ent}$, $LE_{ent}$, $H_{tot}$, and $LE_{tot}$ is strong. The spatial distributions of simulated $H_{sfc}$ and $LE_{sfc}$ are consistent with that of soil moisture. The area with small soil moisture shows large $H_{sfc}$ and small $LE_{sfc}$ because soil moisture plays a critical role in partitioning the available energy into $H_{sfc}$ and $LE_{sfc}$. However, the spatial distributions of $H_{ent}$ and $LE_{ent}$ are not consistent

with that of soil moisture. It seems that the area with strong $H_{sfc}$ has weak $H_{ent}$ and large $LE_{ent}$, which contradicts to the analysis above. A possible explanation can be found in the relationship between entrainment fluxes and cloud liquid water content (QCloud) (Fig. 9). $H_{ent}$ and $LE_{ent}$ at the grid with the maximum sum of QCloud are about 100 and -50 W/m$^2$, respectively. The $H_{ent}$ at the grid with the maximum sum of QCloud is much smaller than that at Site BJ/Nagqu, and the $LE_{ent}$ of the same grid is larger than that at Site BJ/Nagqu. Note that both $H_{ent}$ and $LE_{ent}$ at Site BJ/Nagqu are the entrainment fluxes on a sunny day.

It is reasonable to deduce that the area with large $LE_{ent}$ (over -100 W/m$^2$) and small $H_{ent}$ (below 200 W/m$^2$) is the area with cloud formation. Therefore, high $H_{sfc}$ in the study area is very likely to lead to the convective cloud formation and results in small $H_{ent}$ and large $LE_{ent}$. This is why the spatial distributions of $H_{ent}$ and $LE_{ent}$ are not consistent with that of soil moisture.

### 3.4 Relationship between EF and PBLH

LoCo characteristics using WRF with different combinations of LSMs and PBL schemes can be thoroughly investigated by studying the relationship between the mean daytime EF and maximum daytime PBLH since the PBL growth is a direct response to surface heating. The possible impact of lakes is excluded by dismissing the grids where the mean daytime EF is over 0.9. The relationships between daily mean EF and maximum daytime PBLH by different simulations are shown in Fig. 10.

The relationships between mean daily EF and maximum daytime PBLH in the six simulations are similar but differ in terms

of linear fitting slopes. It seems that the maximum daytime PBLH decreases faster with mean daily EF in the CLM cases than it does in the Noah cases. This is mainly caused by the fact that the mean daytime EF simulated using CLM varies in a much





smaller range, which is much narrower than that simulated using Noah. The slope in the relationship between mean daytime EF and maximum daytime EF simulated using Noah with MYNN is gentler than the slopes simulated by Noah with BouLac and YSU. But the slope simulated CLM with MYNN is steeper than those simulated using CLM with BouLac and YSU. This

can be explained by the frequency distributions of $H_{sfc}$ and $LE_{ent}$ in Fig. 7.

The similarities and differences in the relationships between simulated mean daytime EF and maximum daytime PBLH using the six simulations indicate that the LoCo characteristics simulated by Noah may be more plausible, regardless of which PBL scheme is used.

## 3.5 Sensitivity of LoCo to soil moisture


Sensitivity of LoCo characteristics to different soil moisture conditions in the study area in the rainy season is investigated based on the simulation results using WRF Noah-BouLac with different initial soil moistures. This is because Xu (2018) indicate that the simulation using BouLac produces closest result to the observation, which agrees with the results in this study.

The curves of the mixing diagram at Site BJ/Nagqu with different initial soil moisture conditions are shown in Fig. 11. The

simulated curves of Cp*θ and Lv*q with different initial soil moisture conditions show similar variation but with different magnitude. As the initial soil moisture at Site BJ/Nagqu increases from 0.32 to 0.47 $m^3/m^3$, the simulated $LE_{sfc}$ increases from 220.40 to 245.46 $W/m^2$ and the simulated $H_{sfc}$ decreases from 153.19 to 134.28 $W/m^2$. As a consequence, the simulated Lv*q in the daytime shows a clear gradient as the initial soil moisture increases. Similarly, as the initial soil moisture at Site BJ/Nagqu increases, the simulated $H_{ent}$ decreases from 368.34 to 361.95 $W/m^2$, while the simulated $LE_{ent}$ increases from -222.27 to -

170.63 $W/m^2$. This indicates that as the initial soil moisture increases, less heat and dry air are entrained into the PBL at Site BJ/Nagqu. The results at Site BJ/Nagqu indicate the sensitivity of LoCo to soil moisture under no-cloud conditions because the sum of QCloud in each of the four cases is 0.00 $kg/m^2$ as the initial soil moisture increases.

The frequency distributions of the mean soil moisture at 5 cm and PBL energy budgets in the study area are shown in Fig. 12. As initial soil moisture in the sensitivity experiments increases, the simulated PBL energy budget components change

differently. The simulated $H_{sfc}$ decreases and $LE_{sfc}$ increases as the initial soil moisture increases because soil moisture plays a critical role in the surface available energy partitioning. But the changes in the simulated $H_{ent}$ and $LE_{ent}$ as the initial soil moisture increases are complex. As the initial soil moisture increases, there is an increase in the frequency of $H_{ent}$ ranging from 80 to 240 $W/m^2$ but a decrease in the frequency of $H_{ent}$ below 80 and over 240 $W/m^2$. Similarly, as the initial soil moisture increases, there is an increase in the frequency of $LE_{ent}$ ranging from -240 to -90 $W/m^2$ and a decrease in the frequency of $LE_{ent}$

below -90 and over -240 $W/m^2$. Based on the analysis of the relationship between entrainment fluxes and QCloud in section 3.3, the increases in the frequency of $H_{ent}$ ranging from 80 to 240 $W/m^2$ and $LE_{ent}$ ranging from -240 to -90 $W/m^2$ indicate an increase in convective cloud amount. The relationship of sum of QCloud and entrainment fluxes in different initial soil moisture conditions are shown in Fig. 13. The maximum values of the sum of QCloud in the sensitivity simulations are 0.0192, 0.019, 0.0162, and 0.0150 kg/kg as the initial soil moisture increases. The decrease in the maximum values of the sum of QCloud

may indicate that a decrease in deep cumulus. Therefore, the changes in $H_{ent}$, $LE_{ent}$, and the sum of QCloud may indicate that





an increase in the soil moisture is conducive to an increase in the shallow cumulus amount, but not favorable for the formation of deep cumuli.

As the initial soil moisture increases, the medians of $H_{sfc}$ and $H_{ent}$ decrease from 181.80 to 154.74 W/m$^2$ and from 181.16 to 177.16 W/m$^2$, respectively. The median of $LE_{sfc}$ increases from 169.18 to 201.97 W/m$^2$, while the median of $LE_{ent}$ decreases
from -122.37 to -131.96 W/m$^2$. The median $H_{tot}$ in the study area decreases from 363.91 to 335.95 W/m$^2$. The decrease in $H_{tot}$ is mainly caused by $H_{sfc}$. The median $LE_{tot}$ increase from 44.63 to 66.57 W/m$^2$, and the increase in $LE_{tot}$ is mainly caused by $LE_{sfc}$. It is safe to say that the change in the initial soil moisture has a strong impact on surface fluxes and entrainment fluxes.

The relationship of mean EF and maximum PBLH at daytime in the study area simulated by Noah-BouLac with different initial soil moisture values are shown in Fig. 14. The maximum daytime PBLH decreases as the daytime mean EF increases,
but the slope between maximum PBLH and mean EF at daytime becomes steep as the initial soil moisture increases. This can be attributed to the fact that the number of grids with high EF increases as the initial soil moisture increases and the maximum PBLH decreases due to the increase in shallow convective clouds.

## 4. Discussion

LoCo contains a series of nonlinear processes, which include the interactions among soil states, surface fluxes, PBL development, entrainment, and the formations of convective clouds and precipitations. Accurately modeling these processes is not easy, especially in the TP. In this study, we investigate the LoCo characteristics based on a series of real-case simulations using WRF with different combinations of LSMs and PBL schemes. The simulations using Noah with BouLac, MYNN, and YSU produce better results than those using CLM. The analysis indicates as the initial soil moisture increases from real SM–
0.05 to real SM + 0.1, the domain averaged $H_{sfc}$ decrease from 261.45 to 219.57 W/m$^2$, domain averaged $LE_{sfc}$ increase from 251.93 to 307.51 W/m$^2$. Due to the weakening of surface heating, the max domain averaged PBLH decrease from 2045.0 to 1910.2 m and the max domain averaged convective inhabitation (CIN) increase from 2.5 to 4.23 W/m$^2$. The analysis also reveals the specific influence of surface heating on entrainments and cloud formations at the top of the PBL. It seems that there is a certain critical value for $H_{sfc}$. When $H_{sfc}$ is below the critical value, there is an increase in $H_{ent}$ and a decrease in $LE_{ent}$ as
$H_{sfc}$ increases, which means that more warm and dry air is entrained into the PBL as $H_{sfc}$ increases. When $H_{sfc}$ is over the critical value, there is a decrease in $H_{ent}$ and an increase in $LE_{ent}$ as $H_{sfc}$ increases, which means that less warm and dry air is entrained into the PBL. This is because the formation of convective clouds at the top of the PBL weakens the intensity of entrainments. The is a direct evidence of the impact of the surface heating on the formation of convective clouds.

Another interesting fact is the sensitivity of convective clouds formations to soil moisture in the study area on the sunny
day in the rainy season. The temporal evolution of domain average cloud water content (the sum of ice water and liquid water) under different initial soil moisture conditions indicate that the cloud forms in 11:00 in all the cases and that domain average cloud water content in the real SM - 0.05 case is larger than that in the real SM + 0.1 case until 16:00 (which is not shown here). The analysis show that the simulated QCloud over a dry soil case is larger than that over a wet soil. Therefore, the impact of relatively strong surface heating over a dry soil on the cloud formations is not that straightforward. More observation data





including sunny days, cloudy days and days with convective rain events will be collected in the future. With these data, we will study the LoCo characteristics and the underlying mechanism how the soil state influences the formation of convective clouds and precipitations over a complex surface in rainy season in TP.

## 5. Conclusions

In this study, the LoCo characteristics over a typical underlying surface in the rainy season over the TP were simulated and
discussed using different combinations of LSMs and PBL schemes. The sensitivity of LoCo to soil moisture over a typical underlying surface was investigated via simulations with different initial soil moisture conditions.

The simulated LoCo characteristics using different combinations of LSMs and PBL schemes behave differently in this study. In terms of the curves of mixing diagrams, surface fluxes, and entrainment fluxes, Noah with different PBL schemes produces more realistic curves, $H_{sfc}$, $LE_{sfc}$, and $LE_{ent}$ and smaller $H_{ent}$ than CLM with different PBL schemes does. The frequency
distributions of $H_{sfc}$, $LE_{sfc}$, $H_{ent}$, and $LE_{ent}$ in the study area confirmed the differences in the simulation results. It was also found that the spatial distributions of $H_{sfc}$ and $LE_{sfc}$ in the study area were consistent with that of soil moisture, but the spatial distributions of $H_{ent}$ and $LE_{ent}$ were quite different from that of soil moisture. The reason for the disagreements is that high $H_{sfc}$ may lead to cloud formation, which decreases $H_{ent}$ and increases $LE_{ent}$. The simulations using Noah produce a reasonable relationship between the maximum PBLH and mean daytime EF because the simulations using Noah can produce reasonable
mean daytime EF in the study area.

Sensitivity analysis of LoCo characteristics to soil moisture reveals that the changes in entrainment fluxes and surface fluxes to soil moisture are not the same with soil moisture increase. The changes in entrainment fluxes may be complex due to the possible presence of convective clouds. The results also show that the increase in initial soil moisture may cause an increase in shallow cumuli and a decrease in QCloud

This study investigated the LoCo characteristics on a sunny day over a typical underlying surface in the central TP in terms of the role of the soil moisture in PBL energy growth and convective clouds based on in-situ measurements and numerical simulations. It is the first part of our LoCo analysis in the TP. More observation data including sunny days, cloudy days and days with convective rain events will be collected. We will focus on the basic fact and the possible underlying mechanisms of LoCo over complex surface in TP by exploring the possible influence of the heterogeneity of land surface conditions on LoCo
characteristics.

*Code and data availability.* The simulations presented in this study were conducted using the WRF model, which can be download from its official website (https://www.mmm.ucar.edu/weather-research-and-forecasting-model) free of charge. The forcing data of the ERA-Interim are available at its official website
(https://www.ecmwf.int/en/forecasts/datasets/reanalysis-datasets/era-interim). The simulation results presented in this paper are available from the corresponding author upon request. The observational data at Site BJ/Nagqu are available from Profs. Zeyong HU and Yaoming MA upon request.





*Author contributions.* Genhou SUN conceived the initial idea for this work and completed the simulation analysis and manuscript. Zeyong HU and Yaoming MA produced the valuable observation data for this work. Zhipeng XIE offered valuable suggestions and help the modeling work. Song YANG and Jiemin WANG provided ideas to improve analyses used in this study.

*Competing interests.* The authors declare no conflict of interests.

*Acknowledgments.* The authors thank the National Supercomputer Center in Guangzhou and Zhuhai Joint Innovative Center for Climate, Environment and Ecosystem, China for their support.

*Financial support.* The National Key Research and Development Program of China (2018YFC1505701) and the National Scientific Foundation of China (Grants 41805009, 41675106, 91637208, and 91837208).

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

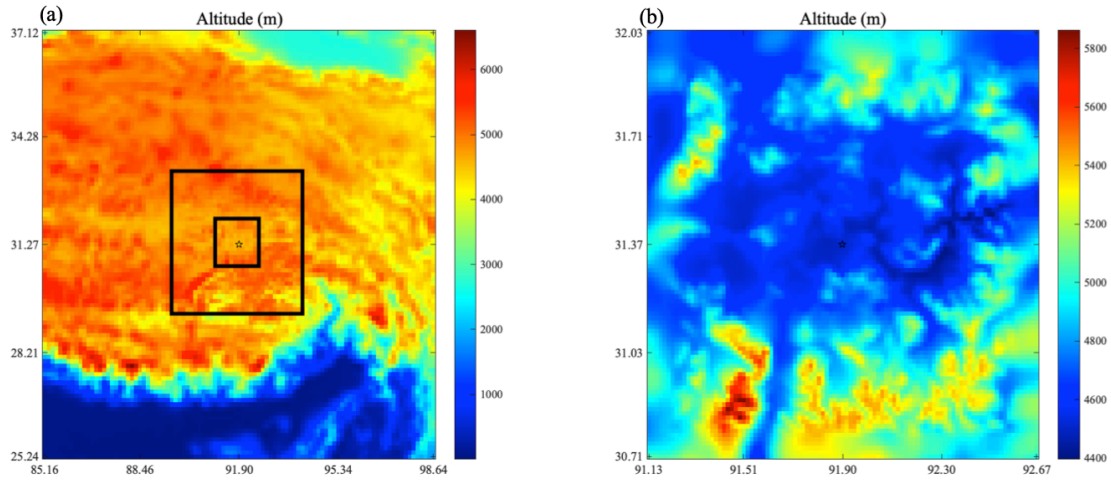

Fig 1 (a) 3-nesting domains in the simulations and (b) Altitude in the study area and the mark (☆) is the location of site

BJ/Nagqu

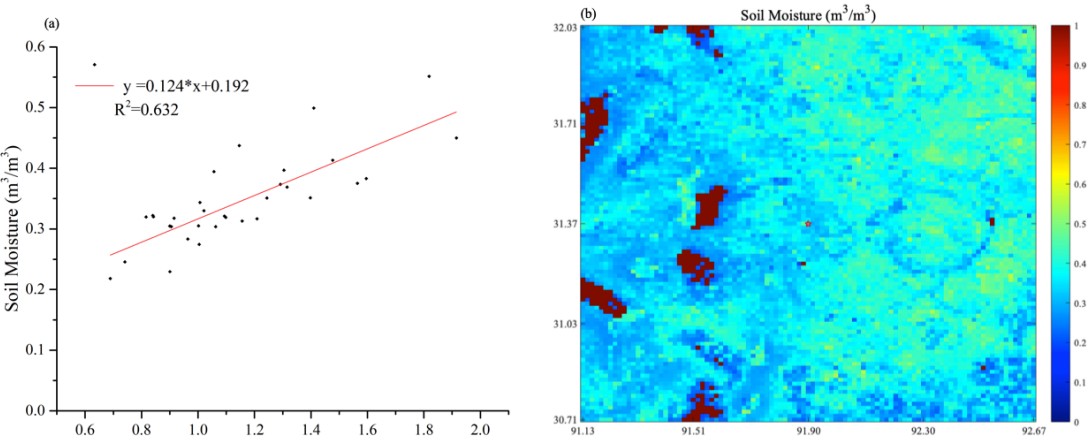

Fig 2 (a) Relationship between LAI from MODIS and the observed soil moisture at 5cm and (b) Modified soil moisture in

the innermost layer of wrfinput. The mark (☆) is the location of site BJ/Nagqu



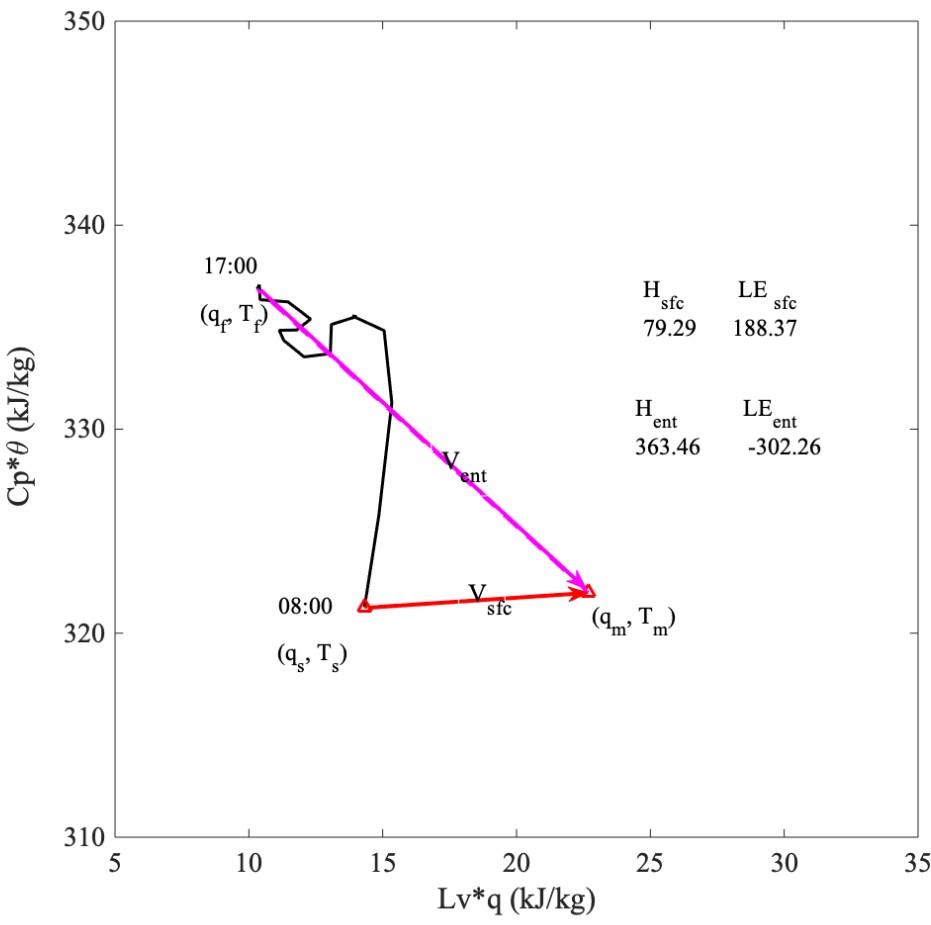

Fig. 3 Mixing diagram analysis based on observation of August 07, 2011



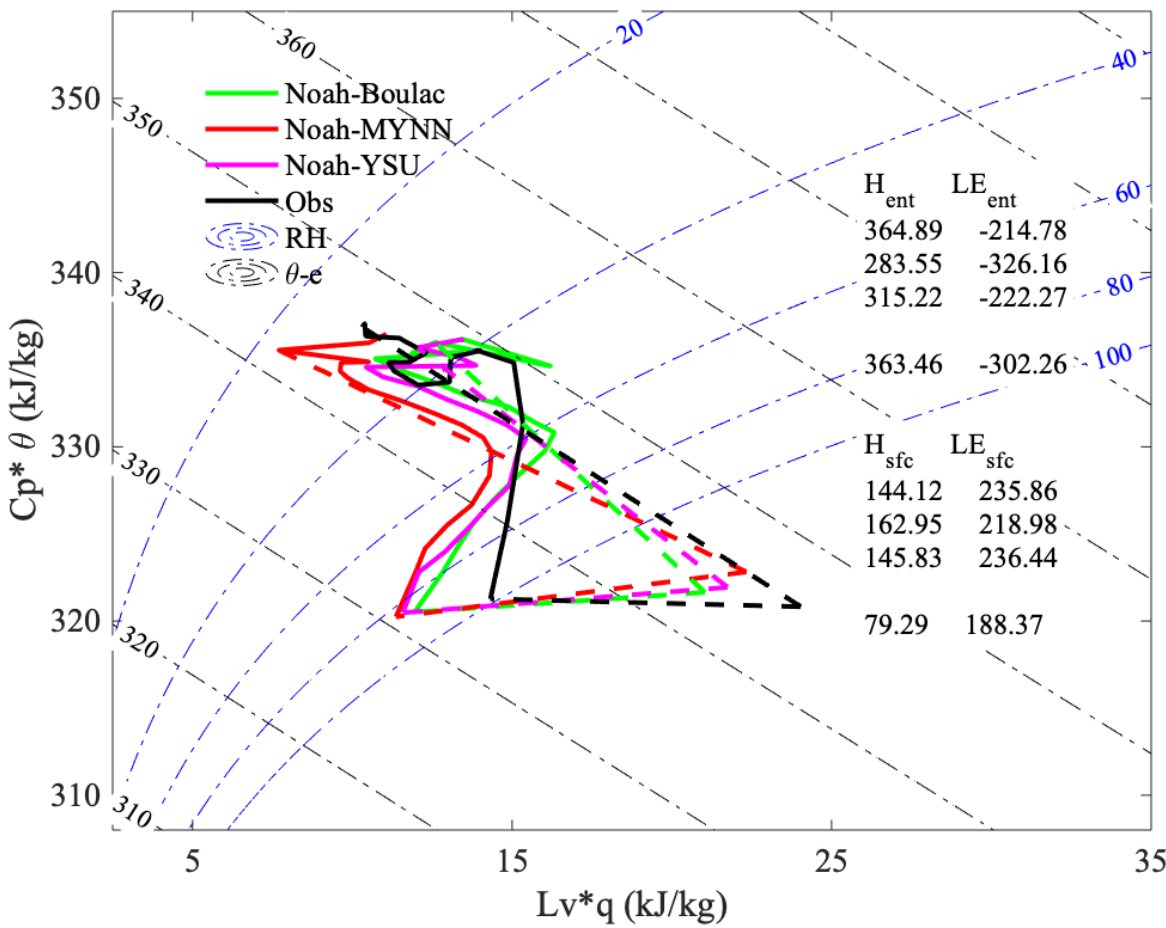

Fig. 4 Mixing diagram at site BJ/Nagqu on Aug 7, 2011simulated using Noah with different PBL schemes

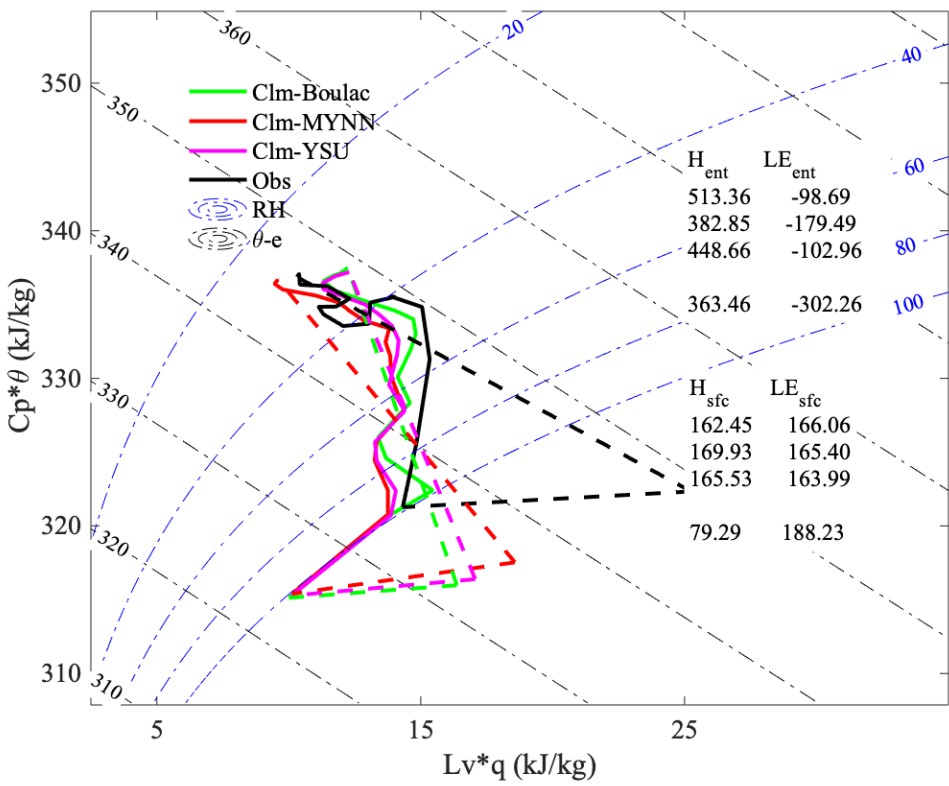

480          Fig. 5 Mixing diagram at site BJ/Nagqu on Aug 7, 2011 simulated using CLM with different PBL schemes





Fig. 6 PBL energy balance at BJ/Nagqu simulated using CLM and Noah with different PBL schemes





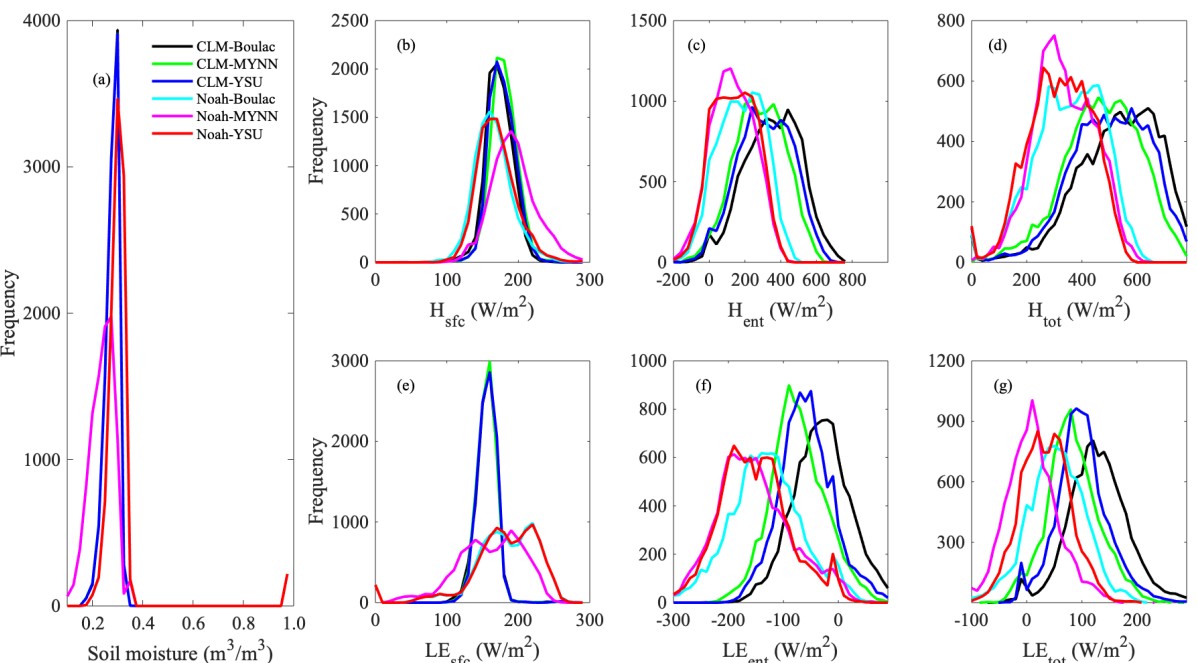

Fig. 7 Frequency distribution of (a) mean soil moisture at 0-10 cm and (b) - (g)PBL energy budgets on August 7, 2011

simulated using different combinations of LSM and PBL schemes

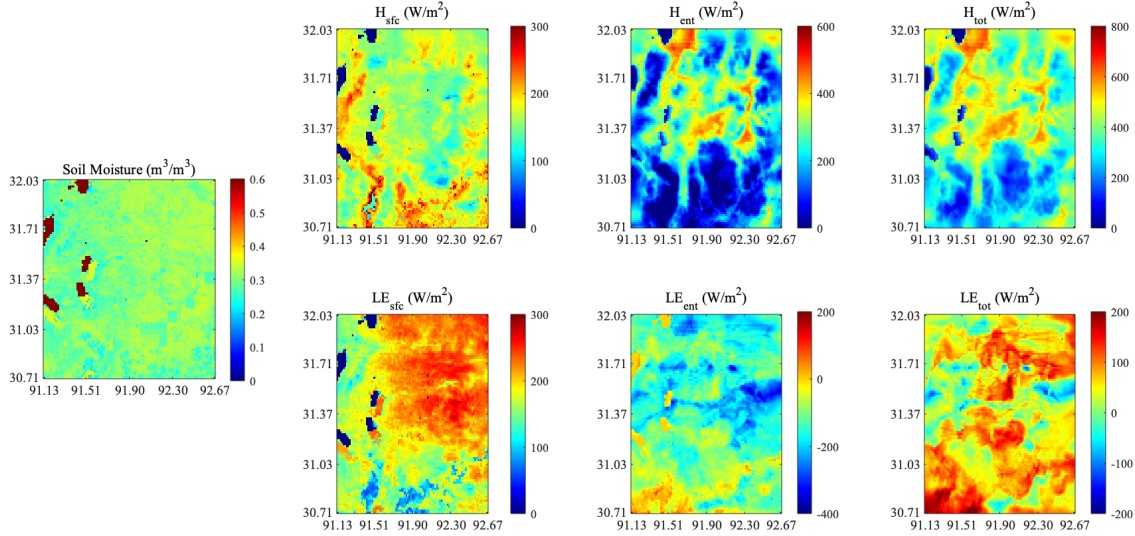

Fig. 8 Spatial distributions of mean soil moisture at 0-10 cm and PBL energy budgets on August 7, 2011 simulated using

WRF with Noah-BouLac. The scale of colorbar for the soil moisture is 0~0.6 in order to highlight the spatial variability of

490                    soil moisture. The soil moisture in the area in dark red is 1.0.



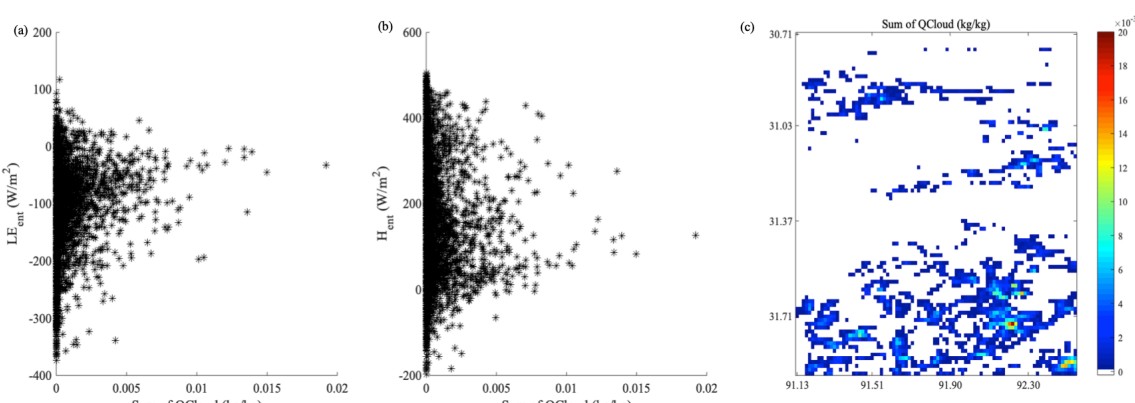

Fig. 9 (a) Relationship between LE$_{ent}$ and sum of QCloud, (b) Relationship between H$_{ent}$ and Qcloud simulated using WRF with Noah-BouLac, (c) spatial distribution of sum of QCloud


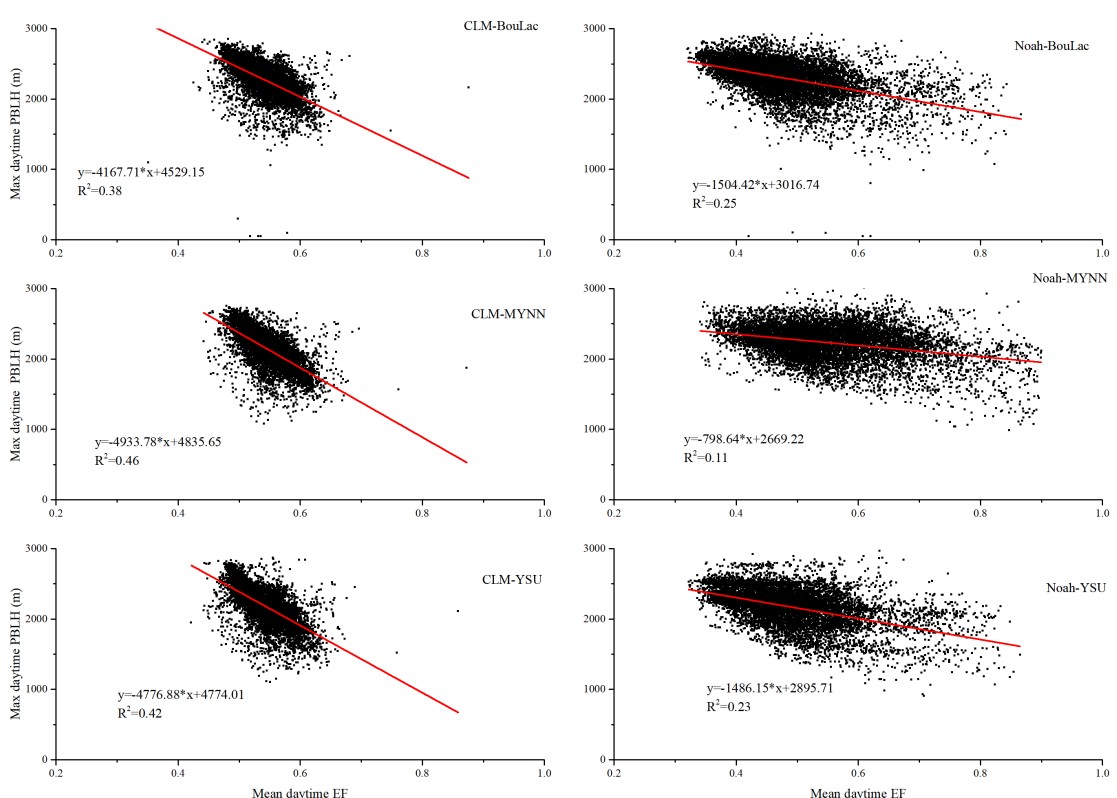

Fig. 10 Relationship between mean daytime EF and the max daytime PBLH simulated by CLM and Noah with different PBL schemes. The grid with mean EF over 0.9 is excluded to avoid the possible influence of lakes in the study area.





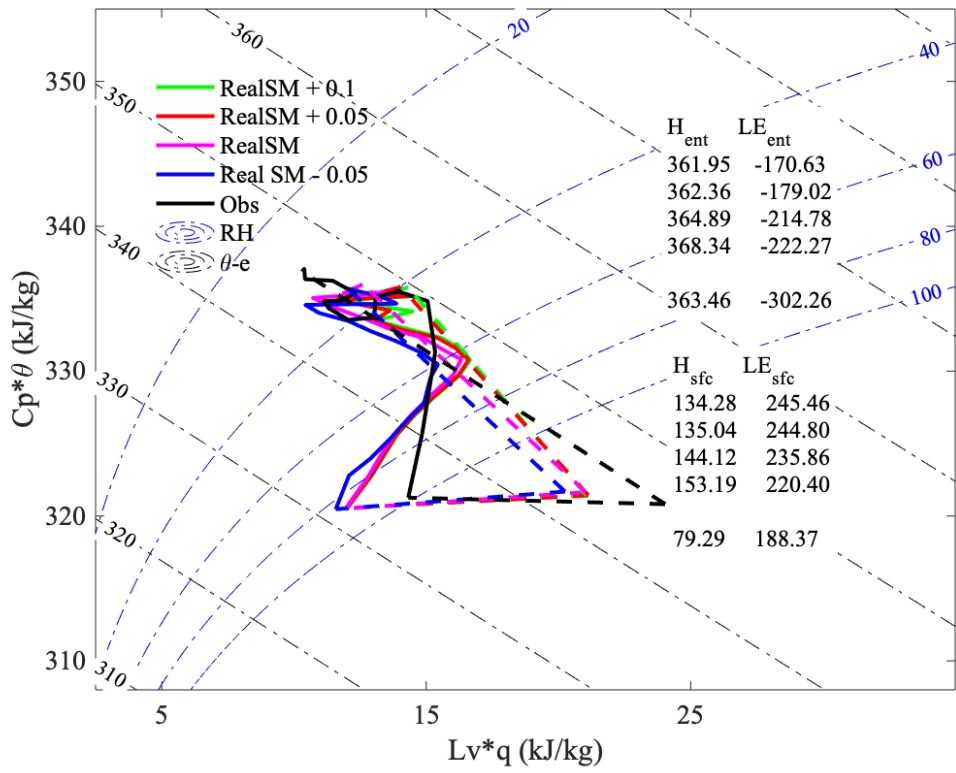

Fig. 11 Mixing diagram under different soil moisture at site BJ/Nagqu simulated using WRF with Noah-BouLac under different initial soil moisture conditions



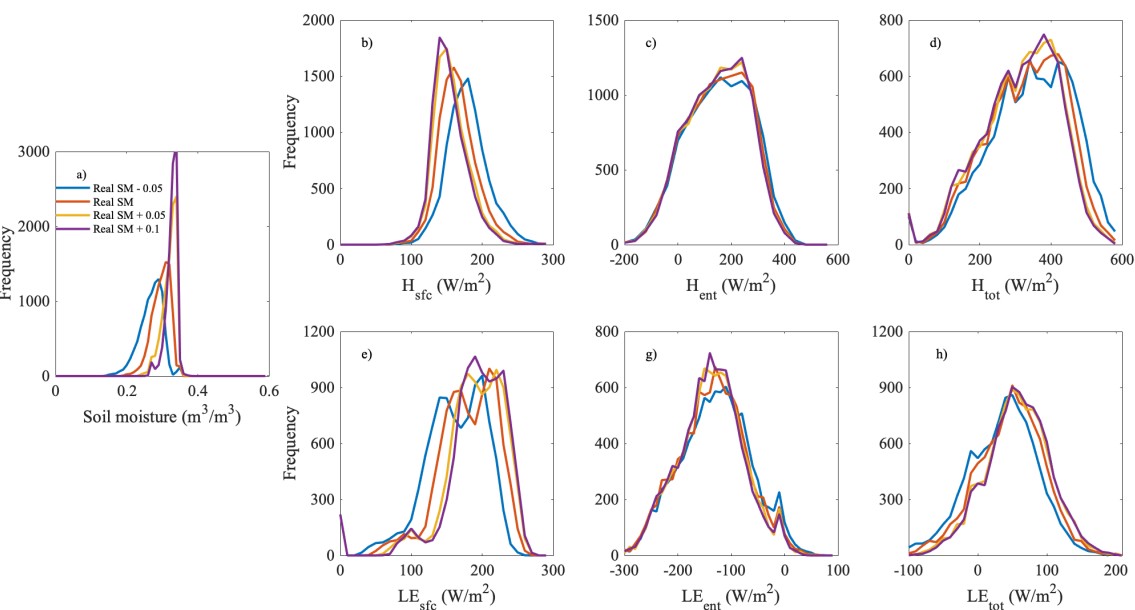

Fig.12 Frequency distribution of (a) mean soil moisture and (b) - (g) PBL energy budget components simulated using Noah-BouLac with different initial soil moisture conditions

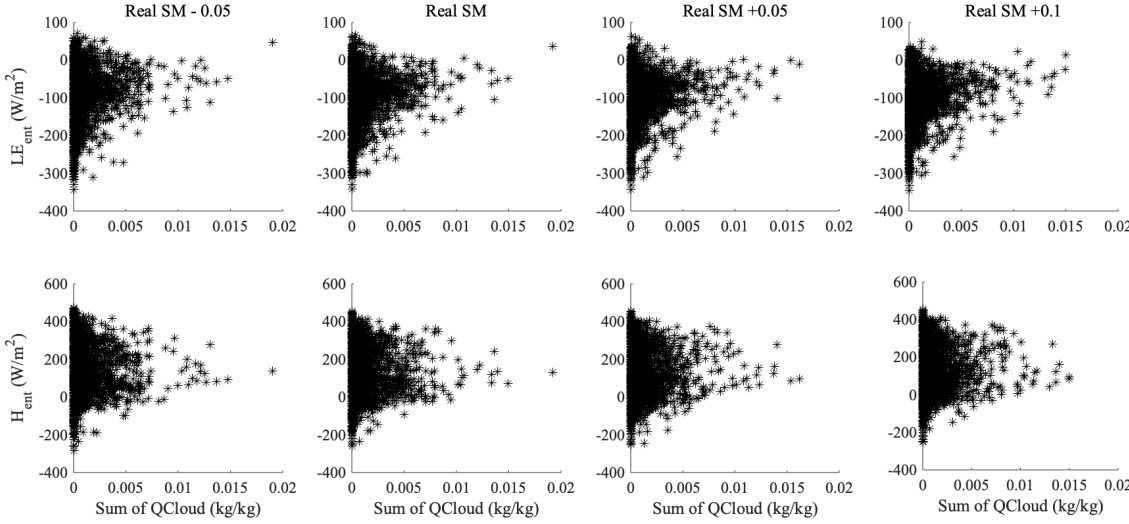

Fig.13 Relationship between sum of Qcloud and entrainment fluxes in the study area simulated using WRF with Noah-BouLac under different initial soil moisture conditions

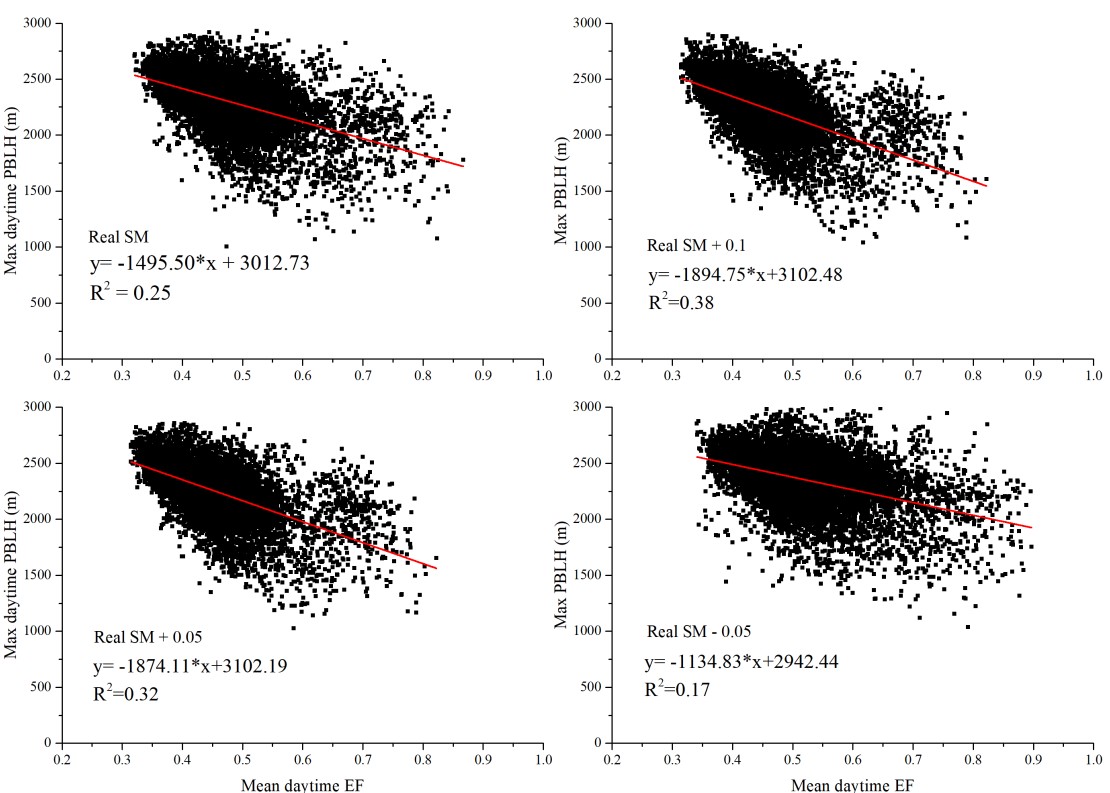

Fig. 14 Relationship between mean daytime EF and the max daytime PBLH simulated using Noah-BouLac with different

initial soil moistures. The grid with mean EF over 0.9 is excluded to avoid the possible influence of lakes in the study area.