# Peer review of "Simulation Analysis of Local Land Atmosphere Coupling in Rainy Season over a Typical Underlying Surface in the Tibetan Plateau"

_Hydrology and Earth System Sciences, 2020_

## Referee Comment (RC1) · Anonymous Referee #1 · 20 Jul 2020

This study investigates the local land-atmosphere coupling over a site in the Tibetan Plateau using the WRF model in a nested domain configuration. The experimental design uses several different LSM and PBL scheme combinations for a 36-hour simulation on a case study day in which many observations exist. The Tibetan Plateau is an area of great interest for water resources priorities, so this type of investigation is quite relevant to a broad community. The results show that the model coupling is sensitivity to the LSM and PBL scheme combination, as well as to the initial soil moisture. These differences ultimately cause changes to convective cloud development.

The paper is generally well-written and the scope is manageable and interesting. That

being said, there are a few areas in which the manuscript should be revised before publication. In particular, the spatial soil moisture map derived from LAI seems to have unreasonably high values of volumetric soil moisture. In addition, although most of the results are discussed appropriately, there are a couple of statements in the results that are not supported by the figures or analyses as they are currently presented. These issues are discussed in more detail below in the major comments.

Major comments:

1) Lines 121-130: The spatial soil moisture map derived from LAI and shown in Figure 2b has a lot of very high values ($>\sim 0.6$). This is worrisome for two reasons. The first is that volumetric soil moisture is not usually that high, so I'm skeptical of how realistic this map is. What are the observed values of soil moisture at the site of interest on this date? What are the min and max soil moisture over the course of the year for this site? Although it is only for one site, this type of analysis could give insight into what is a reasonable max soil moisture for this area. The second potential issue with soil moisture values this high is that they will not even really be used in the Noah LSM. If the authors are using the standard soil texture classifications and soil parameter tables, each soil class has a maximum soil moisture. The maximum soil moisture value varies for each class but even the highest max soil moisture value is less than 0.5. Therefore, all of these high values will essentially be reset to the max SMC for the texture class anyway, making the map more dependent on the soil texture map than on the derived soil moisture from LAI.

2) Section 2.2: Please provide more information on the experimental design. The analysis is 8:00-17:00 local time on August 7, 2011 and the total simulation length is 36-hours, but when is the simulation initialized? Please specify the exact date and time. This is important for understanding the divergence in the starting point in the mixing diagrams between the Noah runs and the CLM runs. The Noah run is wetter and warmer than the CLM run, but they start with the same initial soil moisture and are being forced by the same atmospheric data, right? The amount of time and the time of

day that has passed between the initialization and the figures shown are necessary to understand these differences.

3) Figure 8 does not appear to support the description given of the figure. For example, lines 248-249 say that the soil moisture pattern corresponds with the LEsfc and Hsfc. Except for a couple of spots in the higher elevations in the southwest part of the domain, I don't see how these patterns match up. Perhaps the scale on the figure is not doing the pattern justice? If so, please revise. Otherwise, perhaps the LEsfc corresponds better with the vegetation pattern?

Minor comments:

1) Line 35 describes the Tibetan Plateau as being the Asian Water Tower (i.e., 'also known as'). Line 41, 'TP's role in Asian Water Tower' implies that they are two different things and the TP may affect the Asian Water Tower. Please correct and clarify.

2) Line 98 refers to Fig. 1c, but there is not a Fig. 1c.

3) Lines 129-130: This statement about how the soil moisture was extended to the lower levels is unclear. Was the derived top layer soil moisture used for the entire depth down to 40 cm (so it is uniform vertically)? Please clarify.

4) Line 142: Please specify the exact start date/time that the run was initialized.

5) Lines 148-149: Unless a modification was made to the Noah LSM in this study, the Noah LSM uses a static vegetation dataset. The Noah LSM is the land model for the GFS model. The way the statement is written, it sounds like the Noah LSM is using output from GFS for vegetation, which is not correct. Please revise.

6) Section 3.1: At 8am, the CLM simulations are starting cooler and drier than the Noah simulations. Is this because of the initial conditions at the start of the coupled runs? Is it because of vegetation or other differences in the way CLM and Noah calculate fluxes? This is an important point and should be explored further.

7) Section 3.1: In most of this section, the paper states the statistics and which runs have more or less flux, but there is little explanation of physical explanation behind the statistics. Please consider adding more physical explanation as it would be more interesting to reader if these statistics were translated into what is physically happening in the PBL to cause these differences.

8) Line 263: Does this approach only exclude the water points or also the gridcells nearby water? Why not dismiss the grids where the SM is 1.0 or the land cover is water instead?

9) Lines 266-267: The larger spread in EF seems to imply that there is more surface heterogeneity in Noah than CLM, right? What is the dominant factor causing this? The initial soil moisture is the same for both Noah and CLM, right? Does that mean that it's the treatment of vegetation and/or soil parameters? This should be explored more because it seems to be an important factor of these differences between LSMs.

10) Line 267-268: "...the simulation using BouLac produces closest result to the observation, which agrees with the results in this study." What variables/metrics are being used to determine that Noah-BouLac is the closest to observations for this study? Based on Figure 6, the Noah-BouLac is not the closest to observations. Please clarify and explain.

11) Figures 6 and 7 show the same runs, but the color scheme is different. Please keep a consistent color scheme between these two figures so that it makes it easier on the reader to follow. Also, if possible, please consider reducing the number of colors shown here. You could do that by assigning one color to each PBL scheme (for example orange to YSU) and then using an open icon (i.e., unfilled, just the outline) for CLM and filled icon for the Noah for Figure 6. For Figure 7, you could use one color for each PBL scheme again, but dashed line for CLM and solid for Noah.

Technical corrections:

1) Line 24: conductive should be conducive

2) Line 64: in-sit should be in-situ

3) Line 163: remove 'at the daytime'

4) Line 163: 'furthered this study' -> furthered this 'method' or 'technique' might be more appropriate than study.

---

## Author Comment (AC1) · 31 Jul 2020

Responses to the comments of the Reviewer #1

This study investigates the local land-atmosphere coupling over a site in the Tibetan Plateau using the WRF model in a nested domain configuration. The experimental design uses several different LSM and PBL scheme combinations for a 36-hour simulation on a case study day in which many observations exist. The Tibetan Plateau is an area of great interest for water resources priorities, so this type of investigation is quite relevant to a broad community. The results show that the model coupling is sensitivity to the LSM and PBL scheme combination, as well as to the initial soil moisture. These

differences ultimately cause changes to convective cloud development. The paper is generally well-written and the scope is manageable and interesting. That being said, there are a few areas in which the manuscript should be revised before publication. In particular, the spatial soil moisture map derived from LAI seems to have unreasonably high values of volumetric soil moisture. In addition, although most of the results are discussed appropriately, there are a couple of statements in the results that are not supported by the figures or analyses as they are currently presented. These issues are discussed in more detail below in the major comments. Response: Thanks for your comments and suggestions, which are very helpful to improve our manuscript. We will do our best to answer your precious comments and suggestions and hope our responses would satisfy you.

Major comments: 1ïïjĽ Lines 121-130: The spatial soil moisture map derived from LAI and shown in Figure 2b has a lot of very high values (>âĹij0.6). This is worrisome for two reasons. The first is that volumetric soil moisture is not usually that high, so I'm skeptical of how realistic this map is. What are the observed values of soil moisture at the site of interest on this date? What are the min and max soil moisture over the course of the year for this site? Although it is only for one site, this type of analysis could give insight into what is a reasonable max soil moisture for this area. The second potential issue with soil moisture values this high is that they will not even really be used in the Noah LSM. If the authors are using the standard soil texture classifications and soil parameter tables, each soil class has a maximum soil moisture. The maximum soil moisture value varies for each class but even the highest max soil moisture value is less than 0.5. Therefore, all of these high values will essentially be reset to the max SMC for the texture class anyway, making the map more dependent on the soil texture map than on the derived soil moisture from LAI. Responses: Thanks for your comments. The red area in the Fig. 2b) represents lakes in the study area, where the soil moisture is close to 1.0. The soil moisture in other parts of the study area varies from 0.2 to 0.5. This agrees with the soil parameter tables in Noah LSM. We have added this to the Fig. 2. The interactions between the lakes and land surface
in the domain 3 are simulated in this study. Because this study mainly focuses on the interactions between the land surface and the atmosphere, the simulation result over lakes is not included in the study by discussing the results where the soil moisture is less than 1.0.

2ïijĽ Section 2.2: Please provide more information on the experimental design. The analysis is 8:00-17:00 local time on August 7, 2011 and the total simulation length is 36-hours, but when is the simulation initialized? Please specify the exact date and time. This is important for understanding the divergence in the starting point in the mixing diagrams between the Noah runs and the CLM runs. The Noah run is wetter and warmer than the CLM run, but they start with the same initial soil moisture and are being forced by the same atmospheric data, right? The amount of time and the time of day that has passed between the initialization and the figures shown are necessary to understand these differences. ResponseïijŽ Thanks for your comment. The simulations start in 02:00 August 7, 2011. The simulations from 02:00 to 08:00 of August 7, 2011 are the spin-up time, and the simulations from 08:00 17:00 of August 7, 2011 are used for this analysis. It is true that the initial soil moisture and forced atmospheric data are same for the Noah runs and CLM runs. The simulations of the Noah and CLM runs indicate that there are clear differences in the T2m and q2m and surface fluxes of the Noah and CLM runs, which may be caused by the physics of the Noah and CLM schemes. This has been added to the section 2.2. 3) Figure 8 does not appear to support the description given of the figure. For example, lines 248-249 say that the soil moisture pattern corresponds with the LEsfc and Hsfc. Except for a couple of spots in the higher elevations in the southwest part of the domain, I don't see how these patterns match up. Perhaps the scale on the figure is not doing the pattern justice? If so, please revise. Otherwise, perhaps the LEsfc corresponds better with the vegetation pattern? ResponseïijŽ Thanks for your comments. The spatial distribution of soil moisture in Fig. 8 shows that the soil is dry in the west and south parts of the study area and is generally wet in the middle and east parts of the study area and the areas close to lakes. The Hsfc in the west part of the study area is higher than that in

the east part of the study area except some grids of high altitudes (Fig.1b)). The LEsfc in the east part of the study area is high than that in the west part of the study area. Therefore, it is reasonable to say that the spatial distribution of surface fluxes shows a good agreement of that of soil moisture. For the spatial distribution of surface fluxes, it seems that the altitude in the study area has an influence on the surface fluxes, where the area with high altitudes show high Hsfc and low LEsfc.

Minor comments: 1) Line 35 describes the Tibetan Plateau as being the Asian Water Tower (i.e., 'also known as'). Line 41, 'TP's role in Asian Water Tower' implies that they are two different things and the TP may affect the Asian Water Tower. Please correct and clarify. Response: Thanks for your comments. Tibetan Plateau is known as the Asian Water Tower. The last sentence of paragraph has been rewritten as "Therefore, studying the LoCo over the TP is of great significance for understanding the characteristics of Asian Water Tower". 2) Line 98 refers to Fig. 1c, but there is not a Fig. 1c. Response: Thanks. It is a typo and should be Fig .1b. This typo has been corrected.

3) Lines 129-130: This statement about how the soil moisture was extended to the lower levels is unclear. Was the derived top layer soil moisture used for the entire depth down to 40 cm (so it is uniform vertically)? Please clarify. Response: Thanks for your comment. The variation of soil moisture in the ERA-Interim from 40-cm depth to the top shows very small changes and we assume that the soil moisture from top to 40 cm depth is the same. We thus modified the soil from 40-cm depth to the top by applying the relationship between soil moisture at 5 cm and LAI (Fig. 2 b)). 4) Line 142: Please specify the exact start date/time that the run was initialized. Response: All the simulations start from 02:00 August 7, 2011, Beijing Time, and run for 36 h. The first 6 h of the simulation is for the spin-up, and the simulation results from 08:00 -17:00 August 7, 2011, of the domain 3 are used for the following analysis.

5) Lines 148-149: Unless a modification was made to the Noah LSM in this study, the Noah LSM uses a static vegetation dataset. The Noah LSM is the land model

for the GFS model. The way the statement is written, it sounds like the Noah LSM is using output from GFS for vegetation, which is not correct. Please revise. Response: Thanks for your suggestion. This sentence has been rewritten "A static vegetation dataset based on the monthly Normalized Differential Vegetation Index is used for the Noah LSM." 6) Section 3.1: At 8 am, the CLM simulations are starting cooler and drier than the Noah simulations. Is this because of the initial conditions at the start of the coupled runs? Is it because of vegetation or other differences in the way CLM and Noah calculate fluxes? This is an important point and should be explored further. Response: The CLM and Noah are driven using the same surface conditions (the same initial soil moisture) and atmosphere conditions, and they all start at the same time. Therefore, the most possible reason for the differences in the simulations in the CLM and Noah runs is the differences in the physics of the models.

7) Section 3.1: In most of this section, the paper states the statistics and which runs have more or less flux, but there is little explanation of physical explanation behind the statistics. Please consider adding more physical explanation as it would be more interesting to reader if these statistics were translated into what is physically happening in the PBL to cause these differences. Response: Thanks for your comments. The simulated Hsfc by Noah-MYNN at BJ/Nagqu is larger than those by Noah-BouLac and Noah-YSU while the simulated LEsfc by Noah-MYNN is smaller. This indicates that there are more heat and less vapor into the PBL at BJ/Nagqu in the Noah-MYNN than in the Noah-BouLac and Noah-YSU. According to the Hent and LEent, there is less heat and dryer air entrained into PBL in the Noah-MYNN than that in the Noah-BouLac and Noah-YSU. The differences in Hent and LEent could be attributed to the relatively small PBLH (1418m) by Noah-MYNN than that by Noah-BouLac and Noah-YSU. The simulated Hent and LEent values using CLM with Boulac and YSU are much larger than the observed while, indicating that more heat and less dry air is entrained into PBL than the observed. The Hent using CLM-MYNN is close to the observed while the LEent is larger than the observed, indicating that similar heat and less dry air is entrained into PBL than the observation. We have added these to section 3.1. 8) Line

263: Does this approach only exclude the water points or also the grid cells nearby water? Why not dismiss the grids where the SM is 1.0 or the land cover is water instead? Response: Thanks for your comment, and we accept your suggestion We have checked the result and found that the soil moisture is much a better variable to distinguish the relationship between EF and PBLH over the lake and land. We also found that the simulated soil moisture near the lake is below 0.4, making it very easy to dismiss the lake. The relationships between EF and PBLH in different runs over land are shown as follows.

Fig. 10 Relationship between mean daytime EF and the max daytime PBLH simulated by CLM and Noah with different PBL schemes. The grid in which the mean soil moisture is 1.0 is excluded to avoid the possible influence of lakes in the study 9) Lines 266-267: The larger spread in EF seems to imply that there is more surface heterogeneity in Noah than CLM, right? What is the dominant factor causing this? The initial soil moisture is the same for both Noah and CLM, right? Does that mean that it's the treatment of vegetation and/or soil parameters? This should be explored more because it seems to be an important factor in these differences between LSMs. Response: Thanks for your comment. The larger spread in EF in the Noah run does not imply that there is more heterogeneity in Noah. The frequency distribution of the simulated 5 cm soil moisture of the study area in Fig .7 clearly shows that there is a very small difference in the soil moisture in all the runs. Therefore, the surface heterogeneity in terms of the soil moisture simulated using Noah is only a little more complicated than that simulated using CLM (Fig. 7), and this is not the main reason for the larger spread in EF simulated using Noah. The larger spread in EF simulated using Noah runs is mainly caused by larger variations in Hsfc and LEsfc (Fig. 7) by Noah than those by CLM. According to Fig. 7, the simulated LEsfc in CLM runs vary in narrower ranges than the Hsfc and LEsfc in Noah runs do, while the ranges of Hsfc in CLM runs are similar to those in the Noah runs. This is the main reason for the large spread in EF, which could be attributed to the differences in the performance of CLM and Noah in calculating surface fluxes over a typical underlying surface in Tibetan Plateau. 10)

Line 267-268: "...the simulation using BouLac produces closest result to the observation, which agrees with the results in this study." What variables/metrics are being used to determine that Noah-BouLac is the closest to observations for this study? Based on Figure 6, the Noah-BouLac is not the closest to observations. Please clarify and explain. Response: Thanks for your comment. The frequency distributions of surface fluxes in Fig.7 indicate that the Hsfc and LEsfc in the study area simulated using Noah-BouLac are more acceptable than those using Noah-MYNN. The latter produces larger Hsfc and smaller LEsfc in the study area. The accurate simulation of surface fluxes is very important for the LoCo analysis, and the calculation of entrainment fluxes relies heavily on the surface fluxes. This is why we believe the Noah-MYNN fails to produce reliable surface fluxes, despite Fig. 6 show some supports to Noah-MYNN.

11) Figures 6 and 7 show the same runs, but the color scheme is different. Please keep a consistent color scheme between these two figures so that it makes it easier on the reader to follow. Also, if possible, please consider reducing the number of colors shown here. You could do that by assigning one color to each PBL scheme (for example orange to YSU) and then using an open icon (i.e., unfilled, just the outline) for CLM and filled icon for the Noah for Figure 6. For Figure 7, you could use one color for each PBL scheme again, but dashed line for CLM and solid for Noah. Response: Thanks for your comments. Fig. 6 and 7 have been replotted.

Fig. 6 PBL energy balance at BJ/Nagqu simulated using CLM and Noah with different PBL schemes

Fig. 7 Frequency distribution of (a) mean soil moisture at 0-10 cm and (b) - (g)PBL energy budgets on August 7, 2011, simulated using different combinations of LSM and PBL schemes

Technical corrections: 1) Line 24: conductive should be conducive Response: Thanks. This has been corrected. 2) Line 64: in-sit should be in-situ Response: Thanks. This has been corrected. 3) Line 163: remove 'at the daytime' Response: Thanks. This has

been removed.

4) Line 163: 'furthered this study' -> furthered this 'method' or 'technique' might be more appropriate than study. Response: Thanks. This has been corrected.

Please also note the supplement to this comment:
https://hess.copernicus.org/preprints/hess-2020-199/hess-2020-199-AC1-supplement.pdf

[Figure]

[Figure]

**Fig. 1.** Fig. 8 Spatial distributions of mean soil moisture at 0-10 cm and PBL energy budgets on August 7, 2011 simulated using WRF with Noah-BouLac. The scale of colormap for the soil moisture is 0~0.6 m3/m3

[Figure]

[Figure]

**Fig. 2.** Fig. 10 Relationship between mean daytime EF and the max daytime PBLH simulated by CLM and Noah with different PBL schemes. The grid in which the mean soil moisture is 0.1 is excluded to avoid the pos

**Fig. 3.** Fig. 6 PBL energy balance at BJ/Nagqu simulated using CLM and Noah with different PBL schemes

[Figure]

**Fig. 4.** Fig. 7 Frequency distribution of (a) mean soil moisture at 0-10 cm and (b) - (g)PBL energy budgets on August 7, 2011 simulated using different combinations of LSM and PBL schemes

---

## Referee Comment (RC2) · Anonymous Referee #1 · 1 Sep 2020

Thanks to the reviewers for their responses and changes to the manuscript. Although some of the reviewer's original concerns have been addressed by the revisions, there are still two areas of the manuscript where the reviewer does not agree that the figures support the explanation/text in their current form.

Author Response 1: Thanks for your comments. The red area in the Fig. 2b) represents lakes in the study area, where the soil moisture is close to 1.0. The soil moisture in other parts of the study area varies from 0.2 to 0.5. This agrees with the soil parameter tables in Noah LSM. We have added this to the Fig. 2. The interactions between the lakes and land surface in the domain 3 are simulated in this study. Because this study

mainly focuses on the interactions between the land surface and the atmosphere, the simulation result over lakes is not included in the study by discussing the results where the soil moisture is less than 1.0.

Reviewer Response 1: It is clear that the red area represents lakes. The authors state that the rest of the domain varies from 0.2 to 0.5. This is not shown in the figure. There are plenty of gridcells that are light green and yellow. Given the scale presented in the figure, the case could be made that the darker green gridcells are around 0.5, but the bright green and yellow values are certainly greater than 0.5 and they are plentiful. If it is true that these values are not greater than 0.5, then this figure needs to be revised to convey that information. As it exists now, it does not support the text and my previous comments and concerns remain about this derived map and the impacts of it on the simulations.

Author Response 2: Thanks for your comments. The spatial distribution of soil moisture in Fig. 8 shows that the soil is dry in the west and south parts of the study area and is generally wet in the middle and east parts of the study area and the areas close to lakes. The Hsfc in the west part of the study area is higher than that in the east part of the study area except some grids of high altitudes (Fig.1b)). The LEsfc in the east part of the study area is high than that in the west part of the study area. Therefore, it is reasonable to say that the spatial distribution of surface fluxes shows a good agreement of that of soil moisture. For the spatial distribution of surface fluxes, it seems that the altitude in the study area has an influence on the surface fluxes, where the area with high altitudes show high Hsfc and low LEsfc.

Reviewer Response 2: Regarding "Fig. 8 shows that the soil is dry in the west and south parts of the study area and is generally wet in the middle and east parts of the study are and the areas close to the lakes." The annotated figure showing the areas of low and high soil moisture is appreciated. Although there is slightly more blue (lower SM) in the southern part of the domain as compared to the northern part, it's still very difficult to see any significant groupings of soil moisture. It would likely help to set the

minimum value of the scale/map to a larger value (e.g., 0.2 rather than 0.1) so that more variation can be identified between 0.2 and 0.3 where most of the values are. As it stands, it still looks as if the domain (aside from the lakes) lacks any coherent soil moisture pattern.

―――――――――――――――――――

---

## Author Comment (AC2) · 5 Sep 2020

Thanks for your precious comments and they are very helpful to perfect our manuscript. With your precious comments, we have checked the data and figures carefully and respond to your comments and sincerely hope our responses could answer you're your questions.

Reviewer Response 1: It is clear that the red area represents lakes. The authors state that the rest of the domain varies from 0.2 to 0.5. This is not shown in the figure. There are plenty of gridcells that are light green and yellow. Given the scale presented in the figure, the case could be made that the darker green gridcells are around 0.5, but the

bright green and yellow values are certainly greater than 0.5 and they are plentiful. If it is true that these values are not greater than 0.5, then this figure needs to be revised to convey that information. As it exists now, it does not support the text and my previous comments and concerns remain about this derived map and the impacts of it on the simulations. Response: Thanks for your comments. We have checked the data of Fig. 2 and found that there are 49 grids (0.5% of all the grids in the study area) with the soil moisture ranging from 0.50 to 0.95 m3/m3. The reason for these grids with the soil moisture ranging from 0.50 to 1.0 m3/m3 is caused by the fitting relationship between soil moisture and vegetation index from MODIS in Fig. 2 a. The relationship is applied to the leaf area index (LAI) of MODIS to obtain soil moisture which is more realistic than that obtained from ERA-Interim using the WRF Pre-Processing System (WPS). Luckily, there is a very small percentage of grids with soil moisture over 0.5.

Reviewer Response 2: Regarding "Fig. 8 shows that the soil is dry in the west and south parts of the study area and is generally wet in the middle and east parts of the study are and the areas close to the lakes." The annotated figure showing the areas of low and high soil moisture is appreciated. Although there is slightly more blue (lower SM) in the southern part of the domain as compared to the northern part, it's still very difficult to see any significant groupings of soil moisture. It would likely help to set the minimum value of the scale/map to a larger value (e.g., 0.2 rather than 0.1) so that more variation can be identified between 0.2 and 0.3 where most of the values are. As it stands, it still looks as if the domain (aside from the lakes) lacks any coherent soil moisture pattern. Response: Thanks for your comment. Fig. 8 shows that the soil is dry in the west and south parts of the study area and is generally wet in the middle and east parts of the study are and the areas close to the lakes. Generally, the spatial distributions of mean Hsfc and LEsfc are consistent with that of soil moisture at large scales, although the details of the spatial distribution of mean Hsfc and LEsfc do not agree very well. One possible reason for the weak agreement at small scales is that this is a comparison between the average Hsfc, LEsfc, and soil moisture from 08:00 to 17:00. It is very likely that the averages over 10 hours smooth the details in

the spatial distribution, especially for the Hsfc and LEsfc. This is because the fluxes in the daytime vary significantly due to the daily evolutions of solar radiation as well as the presence of clouds, while the soil moisture in the study area shows very small changes. Therefore, the details of the spatial variability in the Hsfc and LEsfc are very likely to be smoothed in the studied area, leading to the fact that the details of the spatial distribution of mean Hsfc and LEsfc do not agree very well with that of the soil moisture at small scales, as shown in Fig. 8. We have tried to change the range of the scale of soil moisture in Fig.8 and it seem that the range from 0.1 to 0.5 could represent the details of the spatial distribution of the simulated soil moisture, which is shown below.

Please also note the supplement to this comment:
https://hess.copernicus.org/preprints/hess-2020-199/hess-2020-199-AC2-supplement.pdf

—————————————————

[Figure]

Fig. 1. Frequency distribution of soil moisture of the study area. There are only 49 grids with soil moisture ranging from 0.5 to 0.95, which are mainly caused by the linear fitting relationship in Fig. 2 a i

[Figure]

**Fig. 2.** Fig. 8 Spatial distribution of mean soil moisture and PBL energy budgets simulated using WRF with Noah-BouLac

---

## Referee Comment (RC3) · Anonymous Referee #2 · 6 Sep 2020

The author analyzed the local land-atmosphere interaction in the Tibetan Plateau by the aid of regional climate model (WRF) and different land surface parameterizations. It is well-known that it is important to study the planetary boundary processes for the Tibetan Plateau, but the understanding of local land-atmosphere interaction are not enough limited by observations and model's defects in the Tibetan Plateau. The author chose model and parameterizations with good performance validated from in-situ data to further analyze the interactions. The author organized the manuscript well and can be accepted after revisions.

Major comments: 1. The processes happened in planetary boundary are very important, especially for the high altitude regions. Its importance for the Tibetan Plateau has not been well documented in the introduction. Please add some descriptions on this. 2. Previous studies focused on the comparisons of land surface processes from the Noah and CLM. Did you compare them with your results? The authors are suggested to add more discussions by comparing with previous studies. 3. In section 2.2.2, you mentioned several options for PBL schemes in WRF, but you only choose YSU, MYNN, and BouLac parameterizations. Please explain the reason. 4. Figure 6 gives the comparisons among different land surface models and parameterizations. Only from the figure, it is hard to distinguish their different performance. The author can draw conclusions with the help of some quantitative criteria.

Some minor comments: 1. Evapotranspiration, is usually abbreviated as ET, and the author wrote as EF. 2. Lines 37-39, the same to words in lines 11-12 from ABSTRACT, and mentioned again in Lines 42-43. 3. Figure 6, the display of colored label is confused. Different colors represent different schemes, and different marks represent different variables. 4. When mentioned the correlation coefficients, the author should give the significance level, for example for Figure 14.

---

## Author Comment (AC3) · 13 Sep 2020

Responses to the comments of the Reviewer #1

This study investigates the local land-atmosphere coupling over a site in the Tibetan Plateau using the WRF model in a nested domain configuration. The experimental design uses several different LSM and PBL scheme combinations for a 36-hour simulation on a case study day in which many observations exist. The Tibetan Plateau is an area of great interest for water resources priorities, so this type of investigation is quite relevant to a broad community. The results show that the model coupling is sensitivity to the LSM and PBL scheme combination, as well as to the initial soil moisture. These

differences ultimately cause changes to convective cloud development. The paper is generally well-written and the scope is manageable and interesting. That being said, there are a few areas in which the manuscript should be revised before publication. In particular, the spatial soil moisture map derived from LAI seems to have unreasonably high values of volumetric soil moisture. In addition, although most of the results are discussed appropriately, there are a couple of statements in the results that are not supported by the figures or analyses as they are currently presented. These issues are discussed in more detail below in the major comments. Response: Thanks for your comments and suggestions, which are very helpful to improve our manuscript. We will do our best to answer your precious comments and suggestions.

Major comments: 1ïïjĽ Lines 121-130: The spatial soil moisture map derived from LAI and shown in Figure 2b has a lot of very high values (>âĹij0.6). This is worrisome for two reasons. The first is that volumetric soil moisture is not usually that high, so I'm skeptical of how realistic this map is. What are the observed values of soil moisture at the site of interest on this date? What are the min and max soil moisture over the course of the year for this site? Although it is only for one site, this type of analysis could give insight into what is a reasonable max soil moisture for this area. The second potential issue with soil moisture values this high is that they will not even really be used in the Noah LSM. If the authors are using the standard soil texture classifications and soil parameter tables, each soil class has a maximum soil moisture. The maximum soil moisture value varies for each class but even the highest max soil moisture value is less than 0.5. Therefore, all of these high values will essentially be reset to the max SMC for the texture class anyway, making the map more dependent on the soil texture map than on the derived soil moisture from LAI. Responses: Thanks for your comments. The red area in the Fig. 2b) represent lakes in the study area, where the soil moisture is close to 1.0. The soil moisture in other parts of the study area varies from 0.2 to 0.6. There are only 49 grids (0.5% of all the grids in the study area) with the soil moisture ranging from 0.50 to 0.95 m3/m3. The reason for these grids with the soil moisture ranging from 0.50 to 1.0 m3/m3 is caused by the fitting relationship between

soil moisture and vegetation index from MODIS in Fig. 2 a. The relationship is applied to the leaf area index (LAI) of MODIS to obtain soil moisture which is more realistic than that obtained from ERA-Interim using the WRF Pre-Processing System (WPS). Luckily, there is very small percentage of grids with soil moisture over 0.5. Therefore, this still agrees with the soil parameter tables in Noah LSM. We have added this to the Fig. 2. The interactions between the lakes and land surface in the domain 3 are simulated in this study. Because this study mainly focuses on the interactions between the land surface and the atmosphere, the simulation result over lakes is not included in the study by discussing the results where the soil moisture is over 0.9.

2ïïjĽ Section 2.2: Please provide more information on the experimental design. The analysis is 8:00-17:00 local time on August 7, 2011 and the total simulation length is 36-hours, but when is the simulation initialized? Please specify the exact date and time. This is important for understanding the divergence in the starting point in the mixing diagrams between the Noah runs and the CLM runs. The Noah run is wetter and warmer than the CLM run, but they start with the same initial soil moisture and are being forced by the same atmospheric data, right? The amount of time and the time of day that has passed between the initialization and the figures shown are necessary to understand these differences. ResponseïïjŽ Thanks for comments The simulations start in 02:00 August 7, 2011. The simulations from 02:00 to 08:00 of August 7, 2011 are the spin-up time, and the simulations from 08:00 17:00 of August 7, 2011 are used for this analysis. It is true that the initial soil moisture and forced atmospheric data are same for the Noah runs and CLM runs. The simulations of the Noah and CLM runs indicate that there are clear differences in the T2m and q2m and surface fluxes of the Noah and CLM runs, which may be caused by the physics of the Noah and CLM schemes. This has been added to the section 2.2. 3) Figure 8 does not appear to support the description given of the figure. For example, lines 248-249 say that the soil moisture pattern corresponds with the LEsfc and Hsfc. Except for a couple of spots in the higher elevations in the southwest part of the domain, I don't see how these patterns match up. Perhaps the scale on the figure is not doing the pattern

justice? If so, please revise. Otherwise, perhaps the LEsfc corresponds better with the vegetation pattern? ResponseïijŽ Thanks for your comments. The spatial distribution of soil moisture in Fig. 8 show that the soil is dry in the west and south parts of the study area and is generally wet in the middle and east parts of the study area and the areas close to lakes. The Hsfc in the west part of the study area is higher than that in the east part of the study area except some grids of high altitudes (Fig.1b)). The LEsfc in the east part of the study area is high than that in the west part of the study area. Therefore, it is reasonable to say that the the spatial distribution of mean Hsfc and LEsfc are consistent with that of soil moisture at large scales although the details of the spatial distribution of mean Hsfc and LEsfc do not agree very well. One possible reason for the weak agreement at small scales is that this is a comparison between the average Hsfc, LEsfc and soil moisture from 08:00 to 17:00. It is very likely that the averages over 10 hours smooth the details in the spatial distribution, especially for the Hsfc and LEsfc. This is because the fluxes in the daytime vary significantly due to the daily evolutions of solar radiation as well as the presence of clouds, while the soil moisture in the study area shows very small changes. Therefore, the details of the spatial variability in the Hsfc and LEsfc are very likely to be smoothed in the studied area, leading to the fact that the details of the spatial distribution of mean Hsfc and LEsfc do not agree very well with that of the soil moisture at small scales, as shown in Fig. 8.

Fig. 8 Spatial distributions of mean soil moisture at 0-10 cm and PBL energy budgets on August 7, 2011 simulated using WRF with Noah-BouLac. The scale of colormap for the soil moisture is 0∼0.6 m3/m3 in order to highlight the spatial variability of soil moisture. The soil moisture in the area in dark red is 1.0. Minor comments: 1) Line 35 describes the Tibetan Plateau as being the Asian Water Tower (i.e., 'also known as'). Line 41, 'TP's role in Asian Water Tower' implies that they are two different things and the TP may affect the Asian Water Tower. Please correct and clarify. Response: Thanks for your comments. Tibetan Plateau is known as the Asian Water Tower. The last sentence of paragraph has been rewritten as "Therefore, studying the LoCo over

the TP is of great significance for understanding the characteristics of Asian Water Tower". 2) Line 98 refers to Fig. 1c, but there is not a Fig. 1c. Response: Thanks. It is a typo and should be Fig .1b. This typo has been corrected.

3) Lines 129-130: This statement about how the soil moisture was extended to the lower levels is unclear. Was the derived top layer soil moisture used for the entire depth down to 40 cm (so it is uniform vertically)? Please clarify. Response: Thanks for your comment. The variation of soil moisture in the ERA-Interim from 40-cm depth to the top shows very small changes and we assume that the soil moisture from top to 40 cm depth is the same. We thus modified the soil from 40-cm depth to the top by applying the relationship between soil moisture at 5 cm and LAI (Fig. 2 b)). 4) Line 142: Please specify the exact start date/time that the run was initialized. Response: All the simulations start from 02:00 August 7, 2011 Beijing Time, and run for 36 h. The first 6 h of the simulation is for the spin-up, and the simulation results from 08:00 -17:00 August 7, 2011 of the domain 3 are used for the following analysis.

5) Lines 148-149: Unless a modification was made to the Noah LSM in this study, the Noah LSM uses a static vegetation dataset. The Noah LSM is the land model for the GFS model. The way the statement is written, it sounds like the Noah LSM is using output from GFS for vegetation, which is not correct. Please revise. Response: Thanks for your suggestion. This sentence has been rewritten "A static vegetation dataset based on the monthly Normalized Differential Vegetation Index is used for the Noah LSM." 6) Section 3.1: At 8am, the CLM simulations are starting cooler and drier than the Noah simulations. Is this because of the initial conditions at the start of the coupled runs? Is it because of vegetation or other differences in the way CLM and Noah calculate fluxes? This is an important point and should be explored further. Response: The CLM and Noah are driven using the same surface conditions (the same initial soil moisture) and atmosphere conditions, and they all start at the same time. Therefore, the most possible reason for the differences in the simulations in the CLM and Noah runs is the differences in the physics of the models.

7) Section 3.1: In most of this section, the paper states the statistics and which runs have more or less flux, but there is little explanation of physical explanation behind the statistics. Please consider adding more physical explanation as it would be more interesting to reader if these statistics were translated into what is physically happening in the PBL to cause these differences. Response: Thanks for your comments. The simulated Hsfc by Noah-MYNN at BJ/Nagqu is larger than those by Noah-BouLac and Noah-YSU while the simulated LEsfc by Noah-MYNN is smaller. This indicates that there is more heat and less vapor into the PBL at BJ/Nagqu in the Noah-MYNN than in the Noah-BouLac and Noah-YSU. According to the Hent and LEent, there is less heat and dryer air entrained into PBL in the Noah-MYNN than that in the Noah-BouLac and Noah-YSU. The differences in Hent and LEent could be attributed to the relatively small PBLH (1418m) by Noah-MYNN than that by Noah-BouLac and Noah-YSU. The simulated Hent and LEent values using CLM with Boulac and YSU are much larger than the observed while, indicating that more heat and less dry air is entrained into PBL than the observed. The Hent using CLM-MYNN is close to the observed while the LEent is larger than the observed, indicating that similar heat and less dry air is entrained into PBL than the observation. We have added these to section 3.1. 8) Line 263: Does this approach only exclude the water points or also the gridcells nearby water? Why not dismiss the grids where the SM is 1.0 or the land cover is water instead? Response: Thanks for your comment, and we accept your suggestion We have checked the result and found that the soil moisture is much a better variable to distinguish the relationship between EF and PBLH over the lake and land. We also found that the simulated soil moisture near to the lake is below 0.4, making it very easy to dismiss the lake. The relationships between EF and PBLH in different runs over land are shown as follow.

Fig. 10 Relationship between mean daytime EF and the max daytime PBLH simulated by CLM and Noah with different PBL schemes. The grid in which the mean soil moisture is 0.1 is excluded to avoid the possible influence of lakes in the study 9) Lines 266-267: The larger spread in EF seems to imply that there is more surface heterogeneity in Noah than CLM, right? What is the dominant factor causing this?
The initial soil moisture is the same for both Noah and CLM, right? Does that mean that it's the treatment of vegetation and/or soil parameters? This should be explored more because it seems to be an important factor of these differences between LSMs. Response: Thanks for your comment. The larger spread in EF in the Noah run does not imply that there is more heterogeneity in Noah. The frequency distribution of the simulated 5 cm soil moisture of the study area in Fig .7 clearly show that there is very small difference in the soil moisture in all the runs. Therefore, the surface heterogeneity in terms of the soil moisture simulated using Noah is only a little more complicated than that simulated using CLM (Fig. 7), and this is not the main reason for the larger spread in EF simulated using Noah. The larger spread in EF simulated using Noah runs is mainly caused by larger variations in Hsfc and LEsfc (Fig. 7) by Noah than those by CLM. According to Fig. 7, the simulated LEsfc in CLM runs vary in narrower ranges than the Hsfc and LEsfc in Noah runs do, while the ranges of Hsfc in CLM runs are similar to those in the Noah runs. This is the main reason for the large spread in EF, which could be attributed the differences in the performance of CLM and Noah in calculating surface fluxes over a typical underlying surface in Tibetan Plateau. 10) Line 267-268: "...the simulation using BouLac produces closest result to the observation, which agrees with the results in this study." What variables/metrics are being used to determine that Noah-BouLac is the closest to observations for this study? Based on Figure 6, the Noah-BouLac is not the closest to observations. Please clarify and explain. Response: Thanks for your comment. The frequency distributions of surface fluxes in Fig.7 indicate that the Hsfc and LEsfc in the study area simulated using Noah-BouLac are more acceptable than those using Noah-MYNN. The latter produces larger Hsfc and smaller LEsfc in the study area. The accurate simulation of surface fluxes is very important for the LoCo analysis, and the calculation of entrainment fluxes relies heavily on the surface fluxes. This is why we believe the Noah-MYNN fails to produce reliable surface fluxes, despite the Fig. 6 show some supports to Noah-MYNN.

11) Figures 6 and 7 show the same runs, but the color scheme is different. Please keep a consistent color scheme between these two figures so that it makes it easier

on the reader to follow. Also, if possible, please consider reducing the number of colors shown here. You could do that by assigning one color to each PBL scheme (for example orange to YSU) and then using an open icon (i.e., unfilled, just the outline) for CLM and filled icon for the Noah for Figure 6. For Figure 7, you could use one color for each PBL scheme again, but dashed line for CLM and solid for Noah. Response: Thanks for your comments. Fig. 6 and 7 have been replotted.

Fig. 6 PBL energy balance at BJ/Nagqu simulated using CLM and Noah with different PBL schemes

Fig. 7 Frequency distribution of (a) mean soil moisture at 0-10 cm and (b) - (g)PBL energy budgets on August 7, 2011 simulated using different combinations of LSM and PBL schemes

Technical corrections: 1) Line 24: conductive should be conducive Response: Thanks. This has been corrected. 2) Line 64: in-sit should be in-situ Response: Thanks. This has been corrected. 3) Line 163: remove 'at the daytime' Response: Thanks. This has been removed.

4) Line 163: 'furthered this study' -> furthered this 'method' or 'technique' might be more appropriate than study. Response: Thanks. This has been corrected.

Please also note the supplement to this comment:
https://hess.copernicus.org/preprints/hess-2020-199/hess-2020-199-AC3-supplement.pdf
* * *
[Figure]

[Figure]

**Fig. 1.** Fig. 8 Spatial distributions of mean soil moisture at 0-10 cm and PBL energy budgets on August 7, 2011 simulated using WRF with Noah-BouLac. The scale of colormap for the soil moisture is 0~0.6 m3/m3

[Figure]

**Fig. 2.** Fig. 10

[Figure]

The plot shows H (W/m²) versus LE (W/m²) with the following legend:

Symbol key:
- ▲ Surface fluxes
- ■ Entrainment fluxes
- ★ Total fluxes

Color/marker key:
- ▲ Observation (black)
- ▲ CLM-BouLac (red)
- ▲ CLM-MYNN (green)
- ▲ CLM-YSU (blue)
- △ Noah-BouLac (red)
- △ Noah-MYNN (green)
- △ Noah-YSU (blue)

**Fig. 3.** Fig.6

[Figure]

**Fig. 4.** Fig.7

**Supplement:**

On behalf of all authors, I would like to express our sincerely gratitude to the time and considerations of the editors and two reviewers. Your comments and suggestions are very precious and help us to perfect our manuscript.

**Responses to the comments of the Reviewer #1**

**This study investigates the local land-atmosphere coupling over a site in the Tibetan Plateau using the WRF model in a nested domain configuration. The experimental design uses several different LSM and PBL scheme combinations for a 36-hour simulation on a case study day in which many observations exist. The Tibetan Plateau is an area of great interest for water resources priorities, so this type of investigation is quite relevant to a broad community. The results show that the model coupling is sensitivity to the LSM and PBL scheme combination, as well as to the initial soil moisture. These differences ultimately cause changes to convective cloud development.**

**The paper is generally well-written and the scope is manageable and interesting. That being said, there are a few areas in which the manuscript should be revised before publication. In particular, the spatial soil moisture map derived from LAI seems to have unreasonably high values of volumetric soil moisture. In addition, although most of the results are discussed appropriately, there are a couple of statements in the results that are not supported by the figures or analyses as they are currently presented. These issues are discussed in more detail below in the major comments.**

Response: Thanks for your comments and suggestions, which are very helpful to improve our manuscript. We will do our best to answer your precious comments and suggestions.

Major comments:

1)**Lines 121-130: The spatial soil moisture map derived from LAI and shown in Figure 2b has a lot of very high values (>~0.6). This is worrisome for two reasons. The first is that volumetric soil moisture is not usually that high, so I'm skeptical of how realistic this map is. What are the observed values of soil moisture at the site of interest on this date? What are the min and max soil moisture over the course of the year for this site? Although it is only for one site, this type of analysis could give insight into what is a reasonable max soil moisture for this area. The second potential issue with soil moisture values this high is that they will not even really be used in the Noah LSM. If the authors are using the standard soil texture classifications and soil parameter tables, each soil class has a maximum soil moisture. The maximum soil moisture value varies for each class but even the highest max soil moisture value is less than 0.5. Therefore, all of these high values will essentially be reset to the max SMC for the texture class anyway, making the map more dependent on the soil texture map than on the derived soil moisture from LAI.**

**Responses:** Thanks for your comments.

The red area in the Fig. 2b) represent lakes in the study area, where the soil moisture is close to 1.0. The soil moisture in other parts of the study area varies from 0.2 to 0.6. There are only 49 grids (0.5% of all the grids in the study area) with the soil moisture ranging from 0.50 to 0.95 $m^3/m^3$. The reason for these grids with the soil moisture ranging from 0.50 to 1.0 $m^3/m^3$ is caused by the fitting relationship between soil moisture and vegetation index from MODIS in Fig. 2 a. The relationship is applied to the leaf area index (LAI) of MODIS to obtain soil moisture which is more realistic than that obtained from ERA-Interim using the WRF Pre-Processing System (WPS). Luckily, there is very small percentage of grids with soil moisture over 0.5. Therefore, this still agrees with the soil parameter tables in Noah LSM. We have added this to the Fig. 2.

The interactions between the lakes and land surface in the domain 3 are simulated in this study. Because this study mainly focuses on the interactions between the land surface and the atmosphere, the simulation result over lakes is not included in the study by discussing the results where the soil moisture is over 0.9.

2)**Section 2.2: Please provide more information on the experimental design. The analysis is 8:00-17:00 local time on August 7, 2011 and the total simulation length is 36-hours, but when is the simulation initialized? Please specify the exact date and time. This is important for understanding the divergence in the starting point in the mixing diagrams between the Noah runs and the CLM runs. The Noah run is wetter and warmer than the CLM run, but they start with the same initial soil moisture and are being forced by the same atmospheric data, right? The amount of time and the time of day that has passed between the initialization and the figures shown are necessary to understand these differences.**

**Response:** Thanks for comments

The simulations start in 02:00 August 7, 2011. The simulations from 02:00 to 08:00 of August 7, 2011 are the spin-up time, and the simulations from 08:00 17:00 of August 7, 2011 are used for this analysis.

It is true that the initial soil moisture and forced atmospheric data are same for the Noah runs and CLM runs. The simulations of the Noah and CLM runs indicate that there are clear differences in the $T_{2m}$ and $q_{2m}$ and surface fluxes of the Noah and CLM runs, which may be caused by the physics of the Noah and CLM schemes.

This has been added to the section 2.2.

3) **Figure 8 does not appear to support the description given of the figure. For example, lines 248-249 say that the soil moisture pattern corresponds with the LEsfc and Hsfc. Except for a couple of spots in the higher elevations in the southwest part of the domain, I don't see how these patterns match up. Perhaps the scale on the figure is not doing the pattern justice? If so, please revise. Otherwise, perhaps the LEsfc corresponds better with the vegetation pattern?**

**Response:**    Thanks for your comments.

The spatial distribution of soil moisture in Fig. 8 show that the soil is dry in the west and south parts of the study area and is generally wet in the middle and east parts of the study area and the areas close to lakes. The $H_{sfc}$ in the west part of the study area is higher than that in the east part of the study area except some grids of high altitudes (Fig.1b)). The $LE_{sfc}$ in the east part of the study area is high than that in the west part of the study area. Therefore, it is reasonable to say that the the spatial distribution of mean $H_{sfc}$ and $LE_{sfc}$ are consistent with that of soil moisture at large scales although the details of the spatial distribution of mean $H_{sfc}$ and $LE_{sfc}$ do not agree very well. One possible reason for the weak agreement at small scales is that this is a comparison between the average $H_{sfc}$, $LE_{sfc}$ and soil moisture from 08:00 to 17:00. It is very likely that the averages over 10 hours smooth the details in the spatial distribution, especially for the $H_{sfc}$ and $LE_{sfc}$. This is because the fluxes in the daytime vary significantly due to the daily evolutions of solar radiation as well as the presence of clouds, while the soil moisture in the study area shows very small changes. Therefore, the details of the spatial variability in the $H_{sfc}$ and $LE_{sfc}$ are very likely to be smoothed in the studied area, leading to the fact that the details of the spatial distribution of mean $H_{sfc}$ and $LE_{sfc}$ do not agree very well with that of the soil moisture at small scales, as shown in Fig. 8.

[Figure]

Fig. 8 Spatial distributions of mean soil moisture at 0-10 cm and PBL energy budgets on August 7, 2011 simulated using WRF with Noah-BouLac. The scale of colormap for the soil moisture is 0~0.6 m3/m3 in order to highlight the spatial variability of soil moisture. The soil moisture in the area in dark red is 1.0.

Minor comments:

1) **Line 35 describes the Tibetan Plateau as being the Asian Water Tower (i.e., 'also known as'). Line 41, 'TP's role in Asian Water Tower' implies that they are two different things and the TP may affect the Asian Water Tower. Please correct and clarify.**

**Response:** Thanks for your comments.

Tibetan Plateau is known as the Asian Water Tower. The last sentence of paragraph has been rewritten as "Therefore, studying the LoCo over the TP is of great significance for understanding the characteristics of Asian Water Tower".

**2) Line 98 refers to Fig. 1c, but there is not a Fig. 1c.**

**Response:** Thanks. It is a typo and should be Fig .1b. This typo has been corrected.

**3) Lines 129-130: This statement about how the soil moisture was extended to the lower levels is unclear. Was the derived top layer soil moisture used for the entire depth down to 40 cm (so it is uniform vertically)? Please clarify.**

**Response:** Thanks for your comment.

The variation of soil moisture in the ERA-Interim from 40-cm depth to the top shows very small changes and we assume that the soil moisture from top to 40 cm depth is the same. We thus modified the soil from 40-cm depth to the top by applying the relationship between soil moisture at 5 cm and LAI (Fig. 2 b)).

**4) Line 142: Please specify the exact start date/time that the run was initialized.**

**Response:** All the simulations start from 02:00 August 7, 2011 Beijing Time, and run for 36 h. The first 6 h of the simulation is for the spin-up, and the simulation results from 08:00 -17:00 August 7, 2011 of the domain 3 are used for the following analysis.

**5) Lines 148-149: Unless a modification was made to the Noah LSM in this study, the Noah LSM uses a static vegetation dataset. The Noah LSM is the land model for the GFS model. The way the statement is written, it sounds like the Noah LSM is using output from GFS for vegetation, which is not correct. Please revise.**

**Response:** Thanks for your suggestion.

This sentence has been rewritten "A static vegetation dataset based on the monthly Normalized Differential Vegetation Index is used for the Noah LSM."

**6) Section 3.1: At 8am, the CLM simulations are starting cooler and drier than the Noah simulations. Is this because of the initial conditions at the start of the coupled runs? Is it because of vegetation or other differences in the way CLM and Noah calculate fluxes? This is an important point and should be explored further.**

**Response:** The CLM and Noah are driven using the same surface conditions (the same initial soil moisture) and atmosphere conditions, and they all start at the same time. Therefore, the most possible reason for the differences in the simulations in the CLM and Noah runs is the differences in the physics of the models.

7) **Section 3.1: In most of this section, the paper states the statistics and which runs have more or less flux, but there is little explanation of physical explanation behind the statistics. Please consider adding more physical explanation as it would be more interesting to reader if these statistics were translated into what is physically happening in the PBL to cause these differences.**

**Response:** Thanks for your comments.

The simulated $H_{sfc}$ by Noah-MYNN at BJ/Nagqu is larger than those by Noah-BouLac and Noah-YSU while the simulated $LE_{sfc}$ by Noah-MYNN is smaller. This indicates that there is more heat and less vapor into the PBL at BJ/Nagqu in the Noah-MYNN than in the Noah-BouLac and Noah-YSU. According to the $H_{ent}$ and $LE_{ent}$, there is less heat and dryer air entrained into PBL in the Noah-MYNN than that in the Noah-BouLac and Noah-YSU. The differences in $H_{ent}$ and $LE_{ent}$ could be attributed to the relatively small PBLH (1418m) by Noah-MYNN than that by Noah-BouLac and Noah-YSU.

The simulated $H_{ent}$ and $LE_{ent}$ values using CLM with Boulac and YSU are much larger than the observed while, indicating that more heat and less dry air is entrained into PBL than the observed. The $H_{ent}$ using CLM-MYNN is close to the observed while the $LE_{ent}$ is larger than the observed, indicating that similar heat and less dry air is entrained into PBL than the observation.

We have added these to section 3.1.

8) **Line 263: Does this approach only exclude the water points or also the gridcells nearby water? Why not dismiss the grids where the SM is 1.0 or the land cover is water instead?**

**Response:** Thanks for your comment, and we accept your suggestion

We have checked the result and found that the soil moisture is much a better variable to distinguish the relationship between EF and PBLH over the lake and land. We also found that the simulated soil moisture near to the lake is below 0.4, making it very easy to dismiss the lake. The relationships between EF and PBLH in different runs over land are shown as follow.

[Figure]

Fig. 10 Relationship between mean daytime EF and the max daytime PBLH simulated by CLM and Noah with different PBL schemes. The grid in which the mean soil moisture is 0.1 is excluded to avoid the possible influence of lakes in the study

9) **Lines 266-267: The larger spread in EF seems to imply that there is more surface heterogeneity in Noah than CLM, right? What is the dominant factor causing this? The initial soil moisture is the same for both Noah and CLM, right? Does that mean that it's the treatment of vegetation and/or soil parameters? This should be explored more because it seems to be an important factor of these differences between LSMs.**

**Response:**   Thanks for your comment.

The larger spread in EF in the Noah run does not imply that there is more heterogeneity in Noah. The frequency distribution of the simulated 5 cm soil moisture of the study area in Fig .7 clearly show that there is very small difference in the soil moisture in all the runs. Therefore, the surface heterogeneity in terms of the soil moisture simulated using Noah is only a little more complicated than that simulated using CLM (Fig. 7), and this is not the main reason for the larger spread in EF simulated using Noah.

The larger spread in EF simulated using Noah runs is mainly caused by larger variations in $H_{sfc}$ and $LE_{sfc}$ (Fig. 7) by Noah than those by CLM. According to Fig. 7, the simulated $LE_{sfc}$ in CLM runs vary in narrower ranges than the $H_{sfc}$ and $LE_{sfc}$ in Noah runs do, while the ranges of $H_{sfc}$ in CLM runs are similar to those in the Noah runs. This is the main reason for the large spread in EF, which could be attributed the differences in the performance of CLM and Noah in calculating surface fluxes over a typical underlying surface in Tibetan Plateau.

**10) Line 267-268: "...the simulation using BouLac produces closest result to the observation, which agrees with the results in this study." What variables/metrics are being used to determine that Noah-BouLac is the closest to observations for this study? Based on Figure 6, the Noah-BouLac is not the closest to observations. Please clarify and explain.**

**Response:** Thanks for your comment.

The frequency distributions of surface fluxes in Fig.7 indicate that the $H_{sfc}$ and $LE_{sfc}$ in the study area simulated using Noah-BouLac are more acceptable than those using Noah-MYNN. The latter produces larger $H_{sfc}$ and smaller $LE_{sfc}$ in the study area. The accurate simulation of surface fluxes is very important for the LoCo analysis, and the calculation of entrainment fluxes relies heavily on the surface fluxes. This is why we believe the Noah-MYNN fails to produce reliable surface fluxes, despite the Fig. 6 show some supports to Noah-MYNN.

**11) Figures 6 and 7 show the same runs, but the color scheme is different. Please keep a consistent color scheme between these two figures so that it makes it easier on the reader to follow. Also, if possible, please consider reducing the number of colors shown here. You could do that by assigning one color to each PBL scheme (for example orange to YSU) and then using an open icon (i.e., unfilled, just the outline) for CLM and filled icon for the Noah for Figure 6. For Figure 7, you could use one color for each PBL scheme again, but dashed line for CLM and solid for Noah.**

**Response:** Thanks for your comments. Fig. 6 and 7 have been replotted.

[Figure]

Fig. 6 PBL energy balance at BJ/Nagqu simulated using CLM and Noah with different PBL schemes

[Figure]

Fig. 7 Frequency distribution of (a) mean soil moisture at 0-10 cm and (b) - (g)PBL energy budgets on August 7, 2011 simulated using different combinations of LSM and PBL schemes

Technical corrections:

1)   Line 24: conductive should be conducive

Response: Thanks. This has been corrected.

2)   Line 64: in-sit should be in-situ

Response: Thanks. This has been corrected.

3)   Line 163: remove 'at the daytime'

Response: Thanks. This has been removed.

4)   Line 163: 'furthered this study' -> furthered this 'method' or 'technique' might be more appropriate than study.

Response: Thanks. This has been corrected.

**Responses to the comments of the Reviewer #2**

The author analyzed the local land-atmosphere interaction in the Tibetan Plateau by the aid of regional climate model (WRF) and different land surface parameterizations. It is well-known that it is important to study the planetary boundary processes for the Tibetan Plateau, but the understanding of local land-atmosphere interaction is not enough limited by observations and model's defects in the Tibetan Plateau. The author chose model and parameterizations with good performance validated from in-situ data to further analyze the interactions. The author organized the manuscript well and can be accepted after revisions.

Major comments:

1. **The processes happened in planetary boundary are very important, especially for the high-altitude regions. Its importance for the Tibetan Plateau has not been well documented in the introduction. Please add some descriptions on this.**

Respond: Thanks. We have added the following discussion on the previous studies on PBL over TP in the introduction.

The simulation analysis of the PBL over NamCo (Yang et al., 2015) reveals that the Lake Nam Co enhanced the circulation between the lake and land. A study on the reason for the extremely high PBL in the dry season (Chen et al., 2016) reveals that the PBL growth in the dry season is influenced by the surface heating, weak stability of atmosphere and high upper-level potential vorticity. Xu (2018) assessed the performance of eight PBL schemes in producing reliable PBL characteristics over Nagqu area and found that all the PBL schemes produce warm lower-troposphere and higher PBL.

2. **Previous studies focused on the comparisons of land surface processes from the Noah and CLM. Did you compare them with your results? The authors are suggested to add more discussions by comparing with previous studies.**

Respond: Thanks for your comment.

We have compared the simulated $H_{sfc}$ and $LE_{sfc}$ in this study to the three previous studies which focus on the simulating surface fluxes in the central TP in the rainy season. We found that the Noah could produce relatively reliable fluxes while the CLM produce smaller $LE_{sfc}$ in the rainy season.

3. **In section 2.2.2, you mentioned several options for PBL schemes in WRF, but you only choose YSU, MYNN, and BouLac parameterizations. Please explain the reason.**

Response: Thanks for your comments. The reason why we choose the three PBL schemes is that Xu (2018) studied the performance of eight PBL schemes in simulating the PBL thermodynamics in the rainy season and found that the YSU, MYNN and BouLac could produce relatively reliable simulation of PBL thermodynamics. Besides, the YSU is non-local scheme while the MYNN and

BouLac are local ones. The study on the performance of the three schemes could provide valuable information for us.

The reason has been added to the manuscript.

**4. Figure 6 gives the comparisons among different land surface models and parameterizations. Only from the figure, it is hard to distinguish their different performance. The author can draw conclusions with the help of some quantitative criteria.**

**Respond:**    Fig. 6 is the comparison of PBL energy budgets at site BJ among different land surface models and parameterizations. In addition to Fig. 6, the discussions of frequency distributions of PBL energy budget and the relationship between ET and PBLH based on Fig. 7-10 are also the comparison of the simulations using different LSM and PBL schemes.

**Some minor comments:**

1.  **Evapotranspiration, is usually abbreviated as ET, and the author wrote as EF.**

Respond: Thanks. This has been modified.

2.  **Lines 37-39, the same to words in lines 11-12 from ABSTRACT, and mentioned again in Lines 42-43.**

Respond:    Thanks. This has been modified.

3.  **Figure 6, the display of colored label is confused. Different colors represent different schemes, and different marks represent different variables.**

Respond: Thanks. Fig.6 has been modified by assigning one color to each PBL scheme and then using an open icon for CLM and filled icon for the Noah.

4.  **When mentioned the correlation coefficients, the author should give the significance level, for example for Figure 14.**

Respond: Thanks for your suggestion. We have done the t-test for the significance level of the linear regressions in Figs. 10 and 14. The t-test for the regression relationships between mean ET and PBLH in Figs. 10 and 14 show that all the relationships pass the significance level.